# A global Lagrangian eddy dataset based on satellite altimetry

Tongya Liu[1,2] and Ryan Abernathey[3]

[1]State Key Laboratory of Satellite Ocean Environment Dynamics, Second Institute of Oceanography, Ministry of Natural Resources, Hangzhou, China
[2]Southern Marine Science and Engineering Guangdong Laboratory (Zhuhai), Zhuhai, China
[3]Lamont-Doherty Earth Observatory, Columbia University, New York, NY, USA

**Correspondence:** Tongya Liu (liutongya@sio.org.cn)

**Abstract.**

The methods used to identify coherent ocean eddies are either Eulerian or Lagrangian in nature, and nearly all existing eddy dataset are based on the Eulerian method. In this study, millions of Lagrangian particles are advected by satellite-derived surface geostrophic velocities over the period of 1993–2019. Using the method of Lagrangian-averaged vorticity deviation (LAVD), we present a global Lagrangian eddy dataset (GLED v1.0, Liu and Abernathey, 2022, https://doi.org/10.5281/zenodo.7349753 ). This open-source dataset contains not only the general features (eddy center position, equivalent radius, rotation property, etc.) of eddies with lifetimes of 30, 90, and 180 days but also the trajectories of particles trapped by coherent eddies over the lifetime. We present the statistical features of Lagrangian eddies and compare them with those of the most widely used sea surface height (SSH) eddies, focusing on generation sites, size, and propagation speed. A remarkable feature is that Lagrangian eddies is generally smaller than SSH eddies, with a radius ratio of about 0.5. Also, the estimated mass transport by Lagrangian eddies is nearly an order of magnitude smaller than that by the Eulerian calculation, indicating that the coherent contribution to the total eddy transport is very limited. Our eddy dataset provides an additional option for oceanographers to understand the interaction between coherent eddies and other physical or biochemical processes in the Earth system.

## 1 Introduction

Mesoscale eddies, defined as rotating structures ranging typically from tens to hundreds of kilometers and lasting for several weeks to months, are ubiquitous in the global ocean (Fu et al., 2010; Chelton et al., 2011b, hereinafter CS11). And these eddies can trap, transport, and stir tracers such as heat, salt, and biochemical components in the ocean, thereby playing significant roles in nutrient distribution (Chelton et al., 2011a; Frenger et al., 2015), altering large-scale ocean circulation patterns (Abernathey and Marshall, 2013; Liu et al., 2022b), affecting marine ecosystems (Mahadevan, 2016; McGillicuddy Jr, 2016), and modulating climate variability (Busecke and Abernathey, 2019; Li et al., 2022b). Isolated mesoscale eddies in the ocean are generally considered as coherent structures with a material barrier that can trap the fluid within the eddy interior (Haller, 2015). Therefore, understanding the eddy structure and the degree of material transport by eddies are key issues for more accurate parameterization of mesoscale eddies in coarse-resolution marine ecosystem or climate models. To achieve this goal, herein we seek to produce a global coherent eddy dataset based on satellite observations.

Many methods have been proposed to identify mesoscale eddies from numerous oceanic databases such as satellite maps, numerical simulation products, and Argo floats. These existing methods generally fall into two categories: Eulerian and Lagrangian (Haller, 2015; Abernathey and Haller, 2018). The core idea of Eulerian methods is to detect the eddy boundary based on certain physical or geometrical contours from the instantaneous flow field, and then track these boundaries at neighboring times. Frequently used Eulerian eddy boundaries includes contours of Okubo-Weiss parameter, sea surface height (SSH; CS11), potential vorticity (Zhang et al., 2014), velocity streamlines (Nencioli et al., 2010), etc. By contrast, rather than depending on instantaneous images, Lagrangian methods examine trajectories of water parcels over a finite time interval to identify the skeletons of coherent structures. Different techniques such as finite-time Lyapunov exponents (Shadden et al., 2005), finite-scale Lyapunov exponents (d'Ovidio et al., 2009), and Lagrangian-averaged vorticity deviation (LAVD; Haller et al., 2016) have been proposed for eddy detection. Both Eulerian and Lagrangian methods have advantages and disadvantages.

The most significant advantage of Eulerian methods is their operational simplicity: if continuous images of flow fields are available, then searching for eddy centers and boundaries becomes relatively straightforward once the dynamical criterion is determined. This feature means that Eulerian methods are used extensively, especially for SSH eddies (following geostrophic equilibrium) derived from the sea level anomaly (SLA). And the development of satellite observations facilitates eddy identification on a global scale. Using 16 years of altimetry maps with weekly intervals, the first mesoscale eddy dataset was produced (CS11) and the general features of mesoscale eddies were analyzed statistically. Later, Faghmous et al. (2015) presented a global SSH eddy dataset over the period of 1993–2014 using the daily altimetry product and a SLA-based method similar to that used in CS11. Until 2016, the eddy census of CS11 was updated routinely by a research team at Oregon State University, then in 2017 its operation was transferred to CLS/CNES, and it is now distributed by AVISO as the Mesoscale Eddy Trajectory Atlas (META). Several versions of this dataset–from META1.0exp to META3.1exp–are available to users, and Pegliasco et al. (2022) described the improvements from one release to the next. In addition, Dong et al. (2022) constructed a multi-parameter eddy dataset based on the velocity vector field from satellite observations. These Eulerian eddy datasets have been used widely to study the interaction between mesoscale eddies and other processes of the Earth system.

Mesoscale eddies are generally believed to be able to trap and transport the interior fluid when the nonlinearity parameter $U/c$ is greater than 1, where $U$ is the azimuthal eddy speed and $c$ is the eddy propagation speed. Statistics suggest that more than 90% of observed SSH eddies satisfy this criterion (CS11). By assuming no effective water exchange between the eddy interior and background flows, many studies have conducted estimates of heat, salt, and mass transports by Eulerian eddies on regional and global scales (Dong et al., 2014; Zhang et al., 2014; Frenger et al., 2015; He et al., 2018). Among them, the most appealing result shows that the westward zonal eddy mass transport in the subtropical gyre can reach 30–40 Sv, which is surprisingly comparable to the wind-driven gyre transport (Zhang et al., 2014). However, many recent works provide clear evidence that Eulerian methods strongly overestimate the degree of material transport by mesoscale eddies. Horizontally, observations and numerical simulations both suggest that Eulerian eddies are far from coherent structures because there is strong and persistent water exchange across the Eulerian eddy boundary (such as the SSH contour) during the eddy lifespan (Beron-Vera et al., 2013; Wang et al., 2016; Liu et al., 2019, 2022a). The contribution of coherent structures to the total eddy transport is very limited, and most eddy transport is induced by incoherent motions such as swirling and filamentation outside

the eddy cores (Wang et al., 2015; Abernathey and Haller, 2018; Zhang et al., 2019; Xia et al., 2022). In addition, $U/c$ has been shown to be an ineffective indicator of eddy coherent transport because the leakage magnitude of initially trapped water is generally significant and does not depend on this parameter (Liu et al., 2022a). The overestimation of coherent eddy transport might be attributed to the common shortcomings of Eulerian methods (see discussion in Haller, 2015; Abernathey and Haller, 2018). The essential issue is that Eulerian eddy boundaries detected at neighboring times do not necessarily trap the same fluid, and this can be rectified under the Lagrangian framework.

Lagrangian coherent structures have been identified successfully using different techniques. And these eddies can truly trap and transport materials for a certain distance without obvious leakage. However, few studies employ Lagrangian eddies to estimate eddy material transport for the following potential reasons. First, compared with the contour searching of Eulerian methods, Lagrangian algorithms are much more complicated for calculating some physical parameters (e.g., LAVD; details in Section 2) over a time interval. Second, flow fields with high spatial and temporal resolutions are needed to drive millions of Lagrangian particles, which brings huge calculation and storage pressures. Third, the definition method determines that Lagrangian eddies have a preset duration, rather than a free duration like Eulerian eddies, and identifying Lagrangian eddies with different lifetimes is also computationally expensive.

Recently, Abernathey and Haller (2018) used satellite-derived geostrophic velocities in the eastern Pacific to advect Lagrangian particles, and they used the LAVD method to identify rotationally coherent Lagrangian vortices (RCLVs, also called Lagrangian eddies) over a period of 25 years, which is the first large-scale application of objective Lagrangian eddy detection. Based on numerical model outputs, Xia et al. (2022) used the three-dimensional LAVD method to detect global coherent eddies, and they estimated the coherent transport across each latitude or longitude to be only about 1 Sv. Tian et al. (2022) also applied the LAVD method to global eddy detection and presented a 90-day RCLVs dataset, but they adopted a very tight threshold to define the eddy boundary (Tarshish et al., 2018), which would greatly underestimate the size of Lagrangian eddies (see Figure 4).

Nearly all public global eddy datasets are based on the Eulerian framework, and identifying coherent eddies is not an easy task. Therefore, it is necessary to develop a global Lagrangian eddy dataset based on observational data. So far, we have conducted a series of works towards this goal, including regional eddy identification (Abernathey and Haller, 2018; Liu et al., 2022a), parameter sensitivity tests (Tarshish et al., 2018), and numerical experiments (Sinha et al., 2019; Liu et al., 2019; Zhang et al., 2019). In this study, we extend the work of Abernathey and Haller (2018) to the global ocean to identify coherent eddies using the LAVD method, and we generate a Lagragnian eddy dataset based on altimetry observations. This dataset provides not only general features (eddy center position, equivalent radius, rotation property, etc.) of eddies with lifespans of 30, 90, and 180 days but also the trajectory of particles trapped by coherent eddy boundaries over the lifetime, and to the best of our knowledge this is the first attempt at a public eddy dataset. Also, we compare this dataset to the latest SSH eddy dataset (META3.1exp) to understand the statistical differences between the two types of eddies. Our eddy dataset provides an additional option for oceanographers in studying the interactions between coherent eddies and other physical or biochemical processes.

Although some studies have revealed several vertical features of mesoscale eddies, such as regional variability (Zhang et al., 2013), surface and subsurface-intensified types (Dilmahamod et al., 2018), and eddy vertical tilt (Li et al., 2022a), our

understanding of the three-dimensional structure of mesoscale eddies is still limited due to the lack of subsurface observations.
The prevailing assumption is that mesoscale eddies are approximately in geostrophic balance, so this study mainly concentrates
on two-dimensional coherent eddies based on geostrophic currents. We will discuss how unsolved motions affect coherent
eddies later. We encourage users of our product to be mindful of the limitations of the underlying satellite-derived geostrophic
velocity fields used to derive our coherent eddies.
This paper is organized as follows. Section 2 presents the complete process of generating the global Lagrangian eddy dataset.
Section 3 illustrates the basic information of the dataset, the statistical features of coherent eddies, the comparison with SSH
eddies, and the dataset validation. Section 4 introduces the availability of the eddy dataset and related algorithms. Finally,
Section 5 provides the discussion and conclusions.

## 2   Generation of eddy dataset

### 2.1   Satellite altimetry

Because observational data for the subsurface flow field are quite rare, we consider only two-dimensional coherent eddies from
the near-surface geostrophic velocity field $\boldsymbol{v_g} = (u, v)$ that can be derived according to the geostrophic relation

$$\hat{\boldsymbol{k}} \times \boldsymbol{v_g} = -\frac{g}{f}\nabla\eta, \tag{1}$$

where $g$ is the acceleration due to gravity, $f$ is the Coriolis parameter, $\hat{\boldsymbol{k}}$ is the unit vertical vector pointing upward, and $\eta$ is
the SSH. In this study, we use the satellite altimetry product (SEALEVEL_GLO_PHY_L4_REP_OBSERVATIONS_008_047)
distributed by the Copernicus Marine Environment Monitoring Service. This dataset merges along-track measurements from
several altimeter missions and interpolates them to a $1/4°$ latitude-longitude grid. It provides daily variables including the
SLA, the absolute dynamic topography (ADT, equivalent to SSH), and the precomputed geostrophic velocities based on (1).
Note that velocities in the equatorial region (within $\pm 5°$) are estimated based on a higher-order vorticity balance (Lagerloef
et al., 1999) since the geostrophy is not satisfied. We choose the time period of 27 years, from 1 January 1993 to 30 December
2019. In addition, following the procedure described by Abernathey and Marshall (2013), a small correction to the geostrophic
velocities is applied to eliminate the divergence due to the meridional change of $f$ and to perform no-normal-flow boundary
conditions at the coastlines. Compared with noncorrected flow fields, this correction has an insignificant effect on the coher-
ent eddy identification in the open ocean (Abernathey and Haller, 2018). Although the geostrophic current is an incomplete
representation of the full flow in the real ocean, it is by far the leading-order component at the scales of interest in this study.

### 2.2   Particle advection

The first step in generating the global Lagrangian eddy dataset is to advect particles using surface geostrophic velocities (Figure
1). The satellite altimetry product with a $1/4°$ grid resolution can well resolve $\sim 200$ km length structures in the equatorial
region, $\sim 50$ km length structures at the mid-latitudes, and $\sim 25$ km length structures at high latitudes (Ballarotta et al., 2019).

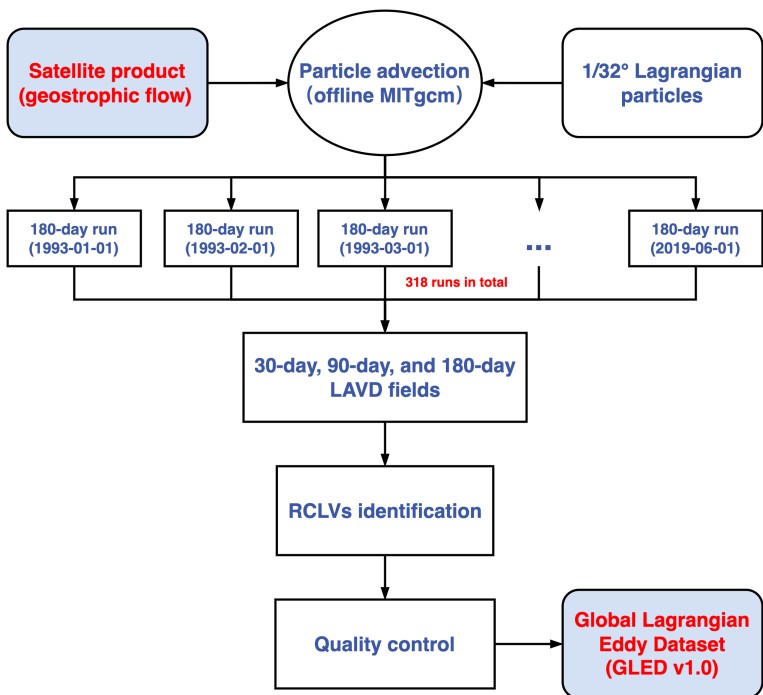

**Figure 1.** Flowchart of eddy dataset generation based on satellite observations.

To reflect properly the fine structure of material transport barriers and Lagrangian eddies, it is necessary to employ an extremely
dense mesh of Lagrangian particles with higher resolution than the forcing velocity field (Haller et al., 2016; Abernathey and
Haller, 2018). However, we should not pursue high resolution particle excessively because of the consequent computational
and storage burdens. Sensitivity tests by Abernathey and Haller (2018) suggest that a particle spacing of $1/32°$ is necessary to
identify RCLVs accurately, and in the present study we use the same resolution and release Lagrangian particles over the global
ocean (between $0°$ and $360°$ longitude and $80°$S and $80°$N latitude; Figure 2a), a total of 39 848 999 points. To our knowledge,
this is the highest resolution to date for a Lagrangian particle mesh applied at global scale. Note that the points on land are
masked because they never move. It is important to note that using ultrahigh-resolution particles does not necessarily improve
the resolution of the flow field as geostrophic currents are inherently unable to resolve small-scale/high-frequency processes,
such as submesoscale flows, tides, and inertia-gravity waves. The real benefit is to avoid the discontinuous areas in the LAVD
fields induced by coarse particle seeding, which allows us to obtain the clear structure of mesoscale coherent eddies.
The MITgcm (Adcroft et al., 2018), an open-source ocean general circulation model, is used to solve the kinematic equation
for Lagrangian particles $d\mathbf{X}/dt = \mathbf{u}$, where $\mathbf{X} = (X, Y)$ is the position vector and $\mathbf{u}$ is a two-dimensional velocity field.
The model can typically operate in either online or offline mode. Here, we employ the offline mode in which the internal
dynamical kernel is turned off and velocity fields are read from preset files with a frequency of 1 day. The FLT package is
enabled to track Lagrangian particles via implementing fourth-order Runge-Kutta integration. Compared with other tools for

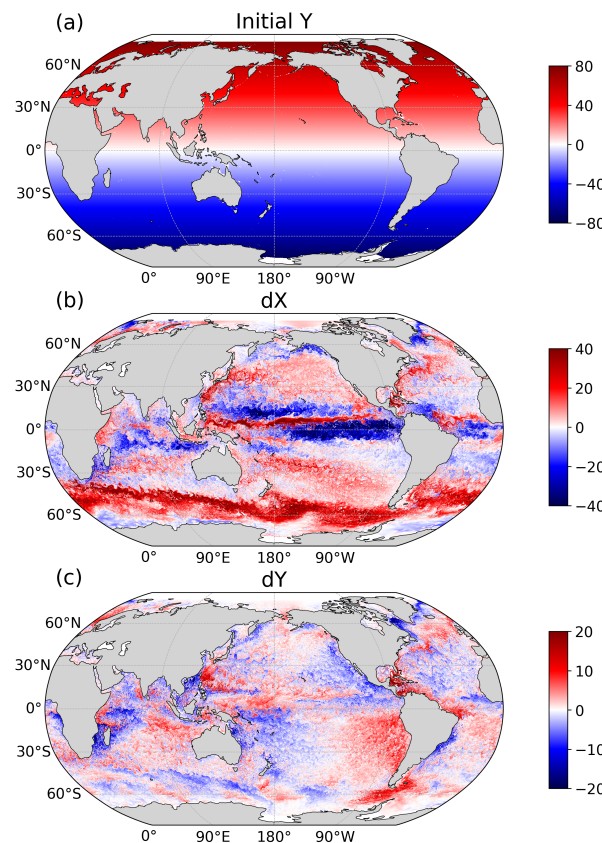

**Figure 2.** (a) Initial latitudes of released Lagrangian particles. (b) Zonal and (c) meridional displacements of particles (in degree) after 180 days.

particle tracking, MITgcm provides a convenient configuration for parallel computing on a high-performance cluster, making the global calculation more efficient.

From January 1993 to June 2019, the Lagrangian particle mesh is initialized on the first day of every month, and these particles are advected forward for 180 days, amounting to 318 180-day runs in total. In the zonal direction, the periodic boundary condition is used to allow particles crossing zero longitude. Figures 2b and 2c show zonal and meridional displacements of particles in a random time interval, which clearly display some main currents (e.g. western boundary currents, zonal tropical currents, and Antarctic Circumpolar Current) and eddy-like structures. In each model run, the relative vorticity is calculated on the Eulerian grid and interpolated to Lagrangian particle positions. To reduce the storage pressure, the relative vorticity and the particle trajectory are output every 10 days, with the total volume still exceeding 20 TB.

## 2.3 Lagrangian eddy identification

Consider a two-dimensional coherent eddy, all fluid parcels along its material boundary should have the same average angular speed when rotating around the eddy core, which is analogous to solid body rotation. Based on this physical intuition, Haller et al. (2016) proposed an objective vorticity-based method to identify the material boundary of a coherent eddy by searching for the outermost closed contour of the LAVD. In a two-dimensional flow, given a finite time interval $(t_0, t_1)$, the LAVD is defined as the average of the vorticity deviation along the Lagrangian particle trajectory, that is,

$$LAVD_{t_0}^{t_1}(x_0, y_0) = \frac{1}{t_1 - t_0} \int_{t_0}^{t_1} |\zeta'[X(x_0, y_0, t), Y(x_0, y_0, t), t]| \, dt, \tag{2}$$

where $(X, Y)$ is the position for the particle released initially at point $(x_0, y_0)$ and $\zeta'$ is the instantaneous relative vorticity deviation from the spatial average over the whole domain. The LAVD (always positive) examines the average magnitude of local rotation for each Lagrangian particle over the time interval. A larger (smaller) LAVD value implies that the particle rotates faster (slower), with the local maximum representing the eddy center and the eddy boundary being the outermost closed LAVD curve encircling the center. This definition determines that all particles inside the boundary must rotate around the eddy core during the time interval, which is essentially different from Eulerian methods based on instantaneous fields.

The algorithm employed for detecting RCLVs has been described in previous studies (Abernathey and Haller, 2018; Tarshish et al., 2018; Liu et al., 2019; Zhang et al., 2019; Liu et al., 2022a). Once a local LAVD maximum is determined, we search outward for closed LAVD curves. There might be multiple closed contours around a center, which are all objective options for the Lagrangian eddy boundary that is expected to be a convex but allowing small deviations. To confine the boundary choice, two parameters are introduced here: the convexity deficiency (CD, Haller et al., 2016) and the coherency index (CI, Tarshish et al., 2018). The CD is defined as the ratio of the area difference between the contour and its convex hull to the total contour's area (see Figure 7 in Tarshish et al., 2018), which means that the closer CD is to zero, the closer the eddy boundary is to being a convex curve. The CI examines the change in spatial compactness of particles inside the contour over a time interval, which is expressed as

$$CI = \frac{\sigma^2(t_0) - \sigma^2(t_1)}{\sigma^2(t_0)}, \tag{3}$$

where $\sigma^2(t) = <|\boldsymbol{X}(t) - <\boldsymbol{X}(t)>|^2>$, $<>$ indicates an average over all particles and $||$ is the standard Euclidean distance. Theoretically, the CI is less than 1 in value, and with decreasing CI, the eddy particle tends to rapidly disperse and develop filaments. The RCLV boundary is determined when the outermost contour satisfies both the CD and CI thresholds.

In this study, the combination of $CD < 0.1$ and $CI > -1$ is adopted according to the sensitivity analysis by Tarshish et al. (2018). Their results indicate that CD values of 0.01, 0.1, and 0.25 are three representative thresholds for strictly coherent, moderately coherent, and leaky vortices, respectively, as is shown in Figure 3a and 3b. Although a small amount of filaments exists, the RCLV defined by $CD < 0.1$ can basically trap the initial water parcels and maintain the coherent structure over the lifetime. It is clear that, the thresholds of 0.25 and 0.01 (adopted by Tian et al., 2022) will greatly overestimate and

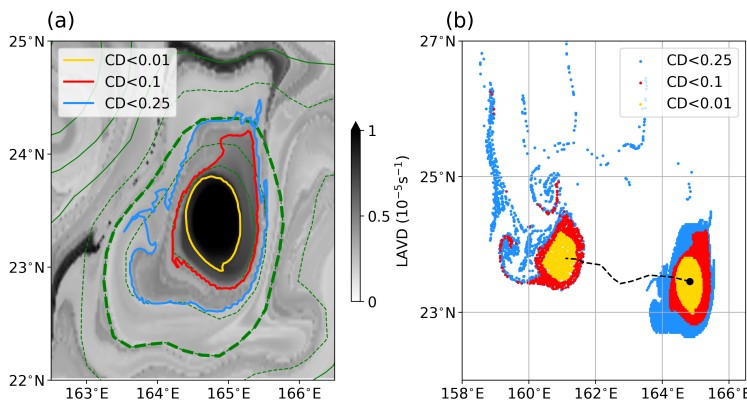

**Figure 3.** (a) A random example of the 30-day LAVD field (color map) and three identified RCLV boundaries using three different CD values. The green dashed thick line is the eddy boundary defined by the SLA contour. (b) Initial and final particle positions trapped by three boundaries. The dashed black line is the eddy center trajectory and the black dot is the initial eddy center.

underestimate, respectively, the size of the coherent eddy. This parameter combination is a moderate threshold for identifying
coherent eddies and has been employed successfully in our previous studies (Liu et al., 2019, 2022a). In addition, we repeatedly
conduct the test of RCLV identification in the random regions and time periods, and as shown in Figure 4, the determined
parameters perform well in identifying RCLVs with lifetimes of 30 and 90 days.
Except for the ability to trap and transport tracers, one of the most significant differences between Eulerian and Lagrangian
eddies is the fact that the LAVD is defined over a specific, fixed finite time interval. Eulerian eddy tracking, in contrast, can
detect eddies of arbitrary lifetimes (of course, without any guarantee of material coherence). Computational pressure dictates
that it is impossible to release Lagrangian particles at any time and identify Lagrangian eddies with an open lifespan, and
to date there is no clear solution to reconcile this difference between the Eulerian and Lagrangian frameworks. In this study,
we choose three typical lifetimes to identify Lagrangian eddies, i.e., 30, 90, and 180 days. Coherent eddies with lifetimes
longer than 180 days are not considered because their number is quite limited based on our results (Figure 6) and those of
Abernathey and Haller (2018). (While eddies of different lifetimes in a specific location may overlap, we cannot say that they
are the "same" eddy because they will, in general, have different material boundaries.) After identifying boundaries for all
eddies over 27 years from 954 LAVD fields, the related eddy parameters (such as radius and movement speed) are calculated,
then we conduct quality control to discard eddies with a radius smaller than 25 km and to check that all the eddy parameters
fall within reasonable ranges. At this point, the Global Lagrangian Eddy Dataset (GLED v1.0, Liu and Abernathey, 2022,
https://doi.org/10.5281/zenodo.7349753 ) has been generated based on satellite observations.

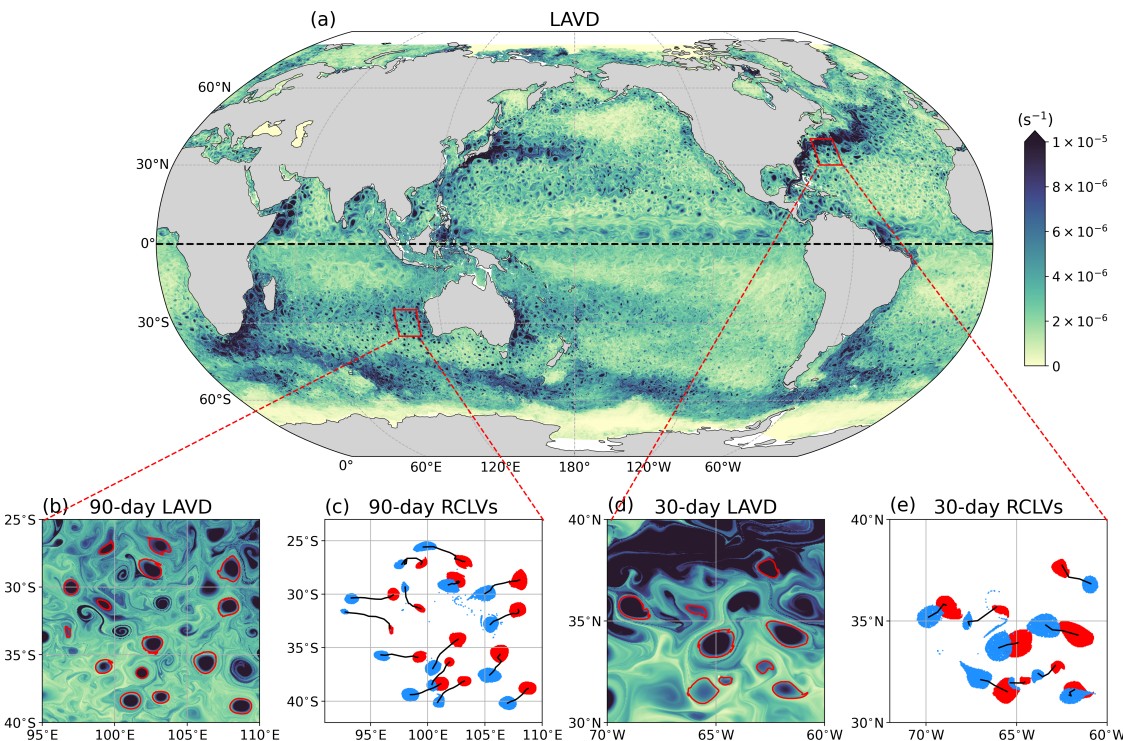

**Figure 4.** (a) 30-day LAVD (in $s^{-1}$) field in the northern hemisphere and 90-day LAVD field in the southern hemisphere calculating from 1 October 2016. (b) Identified boundaries (red contours) of 90-day RCLVs in the west of Australia. (c) Initial (red dots) and final (blue dots) positions of 90-day RCLVs, with black lines representing the eddy center trajectories. (d) Identified boundaries (red contours) of 30-day RCLVs in the Gulf Stream region. (e) Initial (red dots) and final (blue dots) positions of 30-day RCLVs, with black lines representing the eddy center trajectories.

## 3 Results

### 3.1 Description of eddy dataset

GLED v1.0 contains two components. First, the general features of coherent eddies are provided in the directory named *eddyinfo*. The information about 30-day, 90-day, and 180-day eddies is stored separately in three JSON files, which contain the following attributes:

- *id*: an eddy's unique ID composed by identification date, lifetime, and eddy number in the corresponding detection interval;
- *date_start*: generation date of the eddy;
- *duration*: eddy lifespan (in days);
- *radius*: equivalent radius (in kilometers) that is derived from the area enclosed by the eddy boundary;

- *cyc*: eddy rotation type (1 for anticyclonic, -1 for cyclonic);

- *center_lon, center_lat*: the longitude (in degrees North) and latitude (in degrees East) of the eddy center with a frequency of 10 days;

- *dx, dy*: zonal and meridional displacements (in kilometers) of the eddy over the eddy duration;

- *speed_x, speed_y*: averaged zonal and meridional propagation speeds (in meter per second) of the eddy, which equal the displacements divided by the eddy duration;

- *vort*: domain-averaged relative vorticity within the eddy boundary (in per second);

- *lavd*: domain-averaged LAVD value within the eddy boundary (in per second);

Researchers can filter the eddy data based on their studying regions, time periods, or other conditions. For example, if investigating the statistical behaviours of coherent eddies generated around the Kuroshio extension region ($25-35°$N, $140-150°$E), then 2445 30-day, 210 90-day, and 17 180-day eddies over 27 years will be selected for conducting the related analysis.

Second, the trajectories of all Lagrangian particles inside the eddy boundary are provided in the directory named *eddytraj*, which to the best of our knowledge is the first attempt at an open-source eddy dataset. We use an NC file with a three-dimensional array to store the particle positions every 10 days for each eddy, with the array dimensions being particle initial longitude, particle initial latitude, and time. Each NC file is named by its unique eddy ID, and the grid number of the two position dimensions is adjusted according to the eddy size. We randomly load six data records to show the particle positions during the eddy lifetime (Figure 5), and we find that these eddies all perform well in maintaining the coherent structure. An interesting phenomenon is that the eddy in Figure 5a is not initially located around a closed SLA contour, but a coherent structure does exist. This type of coherent eddies are all neglected when using the Eulerian method (Liu et al., 2019). Another typical feature is that the coherent eddy is much smaller than the outermost closed SLA contour (Figure 5b), indicating that this SSH eddy is highly leaky and far from a coherent structure. The second component of GLED v1.0 clearly displays the detailed process of material transport by coherent eddies, which is significant for understanding further the influence of coherent eddies in the distribution of oceanic tracers, especially some biogeochemical tracers such as chlorophyll (Gaube and McGillicuddy Jr, 2017) and nutrients (Hughes and Miller, 2017).

## 3.2 General features of global coherent eddies

To assess GLED v1.0, in this subsection we calculate some statistics of global Lagrangian eddies and compare them with those of a new SSH eddy product (META3.1exp, publicly available at https://www.aviso.altimetry.fr/en/data/products/value-added-products/global-mesoscale-eddy-trajectory-product.html). This dataset updates the detection algorithm and the tracking scheme, and changes the input sea level field from SLA to ADT (Pegliasco et al., 2022), but it is essentially the same as the eddy product proposed by CS11, falling into the Eulerian category.

From January 1993 to December 2019, META3.1exp provides 619 510, 166 426, and 44 329 SSH eddies with radii larger than 25 km and lifetimes longer than 30 days, 90 days, and 180 days, respectively. Our dataset contains many more short-lived

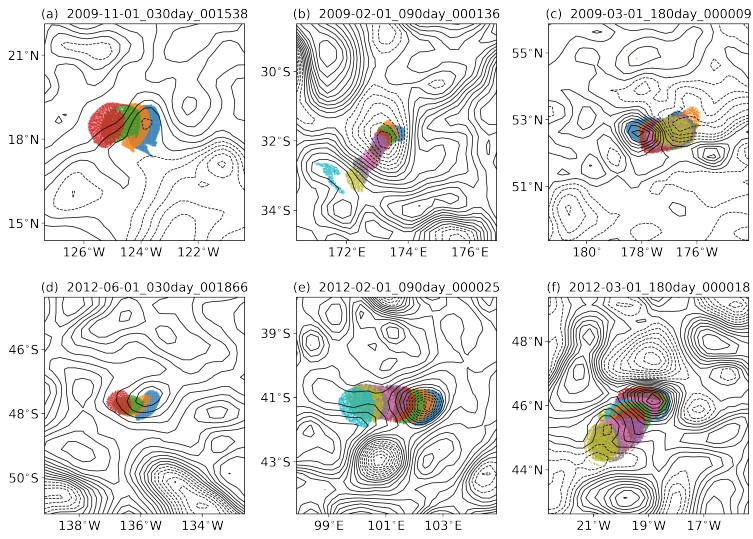

**Figure 5.** Positions of particles (colored dots) inside the eddy boundary every 10 days for randomly selected (a, d) 30-day, (b, e) 90-day, and (c, f) 180-day RCLVs. The unique eddy ID is marked in the title of each panel. Different colors denote the different times, and the blue dots represent the initial positions. The SLA fields are overlaid using black contours with solid lines for positive values and dashed lines for negative values.

but fewer long-lived coherent eddies, with the numbers of 30-, 90-, and 180-day RCLVs in GLED v1.0 being 1 095 356, 116
656, and 13 643, respectively. Census statistics of the numbers for RCLVs and SSH eddies originating in $2° \times 2°$ grids over 27
years are shown in Figure 6. For RCLVs with the three lifetimes, the peak values of eddy number are generally located close to
the eastern boundaries of ocean basins, much higher than that in the western-boundary current regions (Figures 6a, 6c, and 6e).
This spatial feature is not in agreement with the previous analysis by CS11 and the pattern based on META3.1exp, which both
show SSH eddies to be distributed broadly in the mid-latitude regions between 10°N/S and 60°N/S with no obvious east-west
asymmetry (Figures 6b, 6d, and 6f). Compared with SSH eddies with lifetimes longer than 180 days that can be observed nearly
everywhere in the global ocean except for the tropics, the number of 180-day RCLVs is quite limited and they are concentrated
in the southwest of Australia and the interior ocean of the Atlantic.
To understand intuitively the differences between RCLVs and SSH eddies, we choose two regions–one in the northeast
Pacific and the other in the Antarctic Circumpolar Current (ACC)–to display the location and size features of eddies on a
random date (Figure 7). These two regions are selected because they represent weak and strong eddy kinetic energy (EKE)
scenarios. The global EKE map exhibits that the northeast Pacific is less energetic (Whalen et al., 2018) and is typically
considered as a "desert" of long-lived eddies (CS11), but numerous short-lived SSH eddies and RCLVs are distributed widely
(Figure 7a). The most noteworthy feature is that RCLVs are generally smaller in size than SSH eddies and not necessarily
enclosed by the SSH contour. Based on their relative positions to SSH eddies, RCLVs can be classified into two categories (Liu
et al., 2019): overlapping and non-overlapping. The latter are quite different from traditional geostrophic eddies and appears

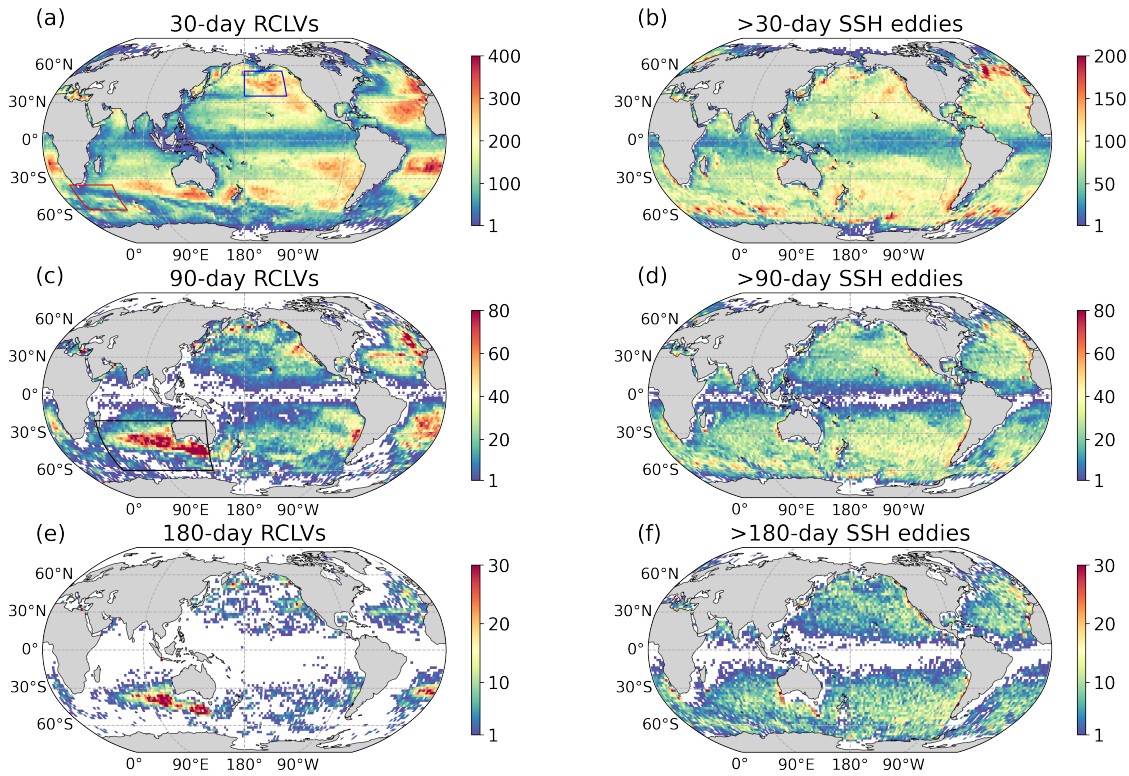

**Figure 6.** The geographic distribution of eddy generation number in $2° \times 2°$ grids for (a, c, e) RCLVs and (b, d, f) SSH eddies over 27 years. Three time intervals (30, 90, and 180 days) are considered and the grid without eddies is masked.

frequently, deserving further investigation of their structure and evolution. Another feature is that many RCLVs propagate eastward over the lifespan in this region, which has not been noticed before. As one of the most energetic regions, the ACC region is rich in SSH eddies with large radii and amplitudes (Figure 7b), but few of them have a coherent core, indicating that these SSH eddies cannot maintain a coherent structure for as little as 30 days. We identify only 39 30-day RCLVs in region 2, much fewer than the number (124) in region 1 with the same size. The reduced number of coherent eddies along the main path of the jet-like current can also be seen clearly in the Gulf Stream and the Kuroshio Extension regions.

We now examine the statistics of eddy radius, zonal propagation speed, and meridional propagation speed for all RCLVs and SSH eddies in $10°$ latitude bins, which are shown using the box plot in Figure 8. Outside of the tropical region, both types of eddies basically decrease in size with latitude, reflecting the dependence of the Rossby deformation radius on the Coriolis parameter (Chelton et al., 1998), but the averaged RCLV radius is only half of the SSH eddy radius, which is consistent with the regional examples shown in Figure 7 and our previous analysis in the eastern Pacific (Abernathey and Haller, 2018). In the tropics, the RCLV radius is only about 40 km because there are numerous non-overlapping RCLVs with small size (not shown). In addition, it is observed that RCLVs and SSH eddies have similar westward propagation speeds, consistent with the

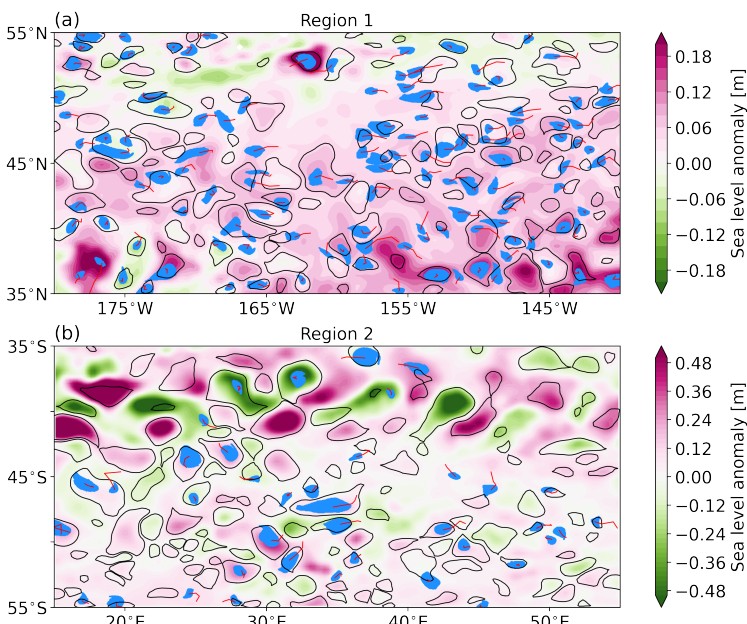

**Figure 7.** Locations of 30-day RCLVs (blue dots) and SSH eddies(>30 days, black contours) in (a) region 1 (blue box in Figure 6a) and (b) region 2 (red box in Figure 6a) on 1 January 2009. The red lines are the center trajectories of the RCLVs, and the color map represents the SLA field.

phase speed of long Rossby wave (Killworth et al., 1997), except for the tropical region where some RCLVs move eastward with the background tropical flows. For the meridional propagation speed, its magnitude is usually lower an order than that of the zonal speed, and both types of eddies have similar patterns, with the difference emerging in $30°S–0°$ where there are many RCLVs along the eastern boundary (see Figure 6a).

### 3.3 Global mass transport by coherent eddies

One application of this eddy dataset is to estimating the mass transport by coherent eddies. Following the methods used by Dong et al. (2014) and Zhang et al. (2014), we calculate the averaged zonal and meridional transport across the section for each $1° \times 1°$ grid. Assuming that the water masses are coherently trapped during the eddy lifespan, the instantaneous zonal transport induced by an individual eddy can be expressed as $V_e C_x$, where $V_e$ is the eddy volume and $C_x$ is the zonal propagation speed. We integrate all eddy snapshots (daily) over the studying period (1993-2019) to obtain the total transport $\sum V_e C_x$ within each bin. Dividing it by the number of satellite snapshots $N$ and the length of one longitude degree $L_x$, we can get the average zonal transport $Q_x$ (in Sverdrup) across the latitude section for each $1° \times 1°$ grid,

$$Q_x = \frac{\sum V_e C_x}{N L_x}. \tag{4}$$

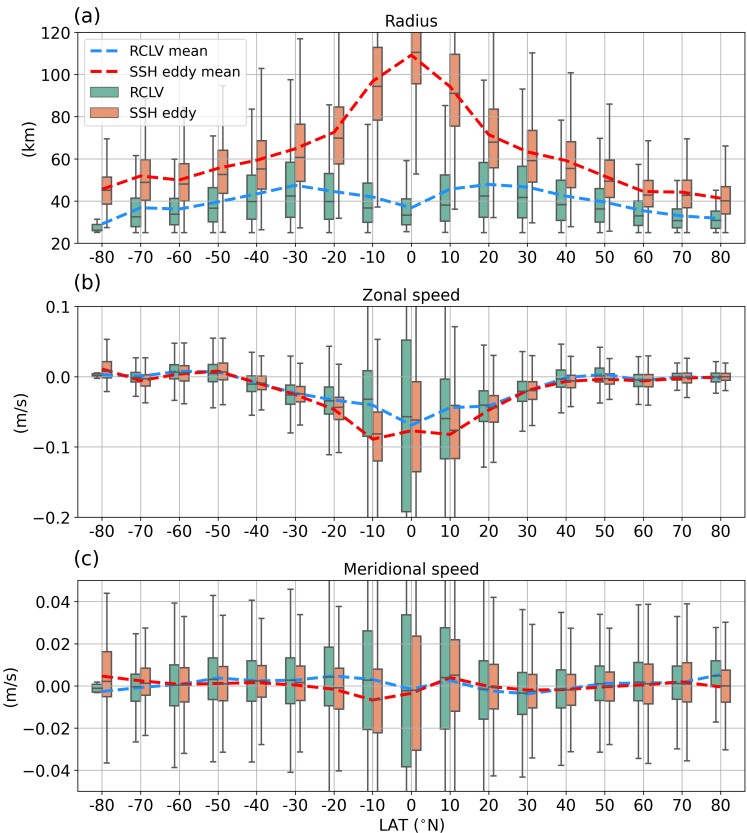

**Figure 8.** Statistics of (a) radius, (b) zonal propagation speed, and (c) meridional propagation speed for RCLVs and SSH eddies. The box plot shows statistics of all eddies in $10°$ bins. The box and the black whisker span the 25th to 75th and 10th to 90th percentiles of the distribution, respectively. The black line in the box indicates the median. The means of all eddies in a bin are shown using dashed lines, blue for RCLVs and red for SSH eddies.

Similarly, the meridional transport $Q_y$ by coherent eddies across the longitude section within the same bin can be obtained,
$$Q_y = \frac{\sum V_e C_y}{N L_y},$$
(5)

where $C_y$ is the meridional propagation speed and $L_y$ is the length of one latitude degree. The eddy volume is calculated by
$V_e = s\pi R^2 h$, where $R$ is the eddy radius, $s = 0.5$ is a correction factor for the eddy vertical structure from Dong et al. (2014),
and $h = 500$ m is the eddy depth. Due to the vertical structure of coherent eddies remaining an open issue, we use a constant
depth following Abernathey and Haller (2018), which is supported by the findings of Roemmich and Gilson (2001) and Xia
et al. (2022). This is a relatively crude estimate with accuracy within an order of magnitude, and the focus here is mainly on
the *difference* between the two types of eddies. Here, 30-day RCLVs and SSH eddies with lifetimes longer than 30 days are
considered from 1993 to 2019.

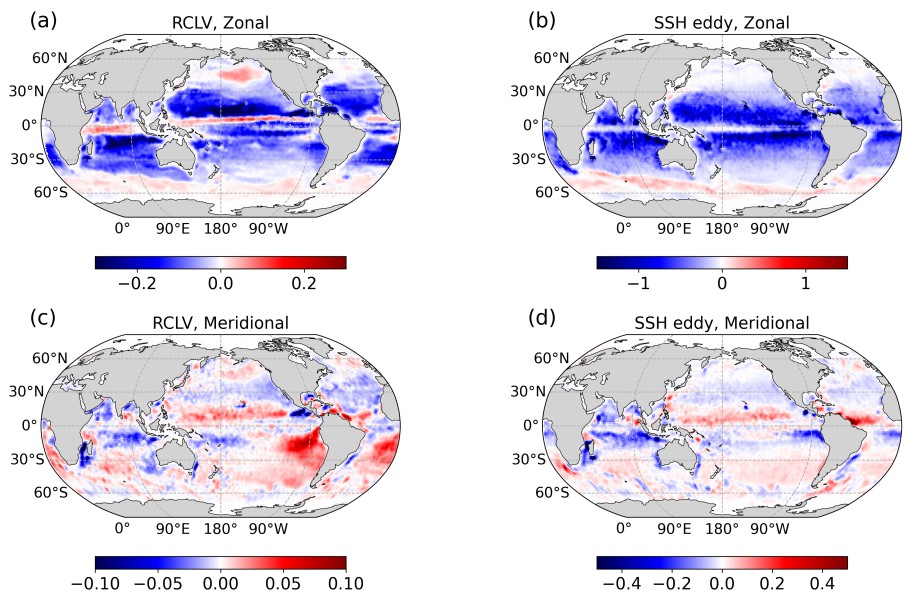

**Figure 9.** Global distributions of the (a, b) zonal and (c, d) meridional transport by (a, c) RCLVs and (b, d) SSH eddies in $1° \times 1°$ bins. Note that different colorbar ranges are used here and the transport unit is the sverdrup [Sv].

Figure 9 shows the global distribution of the zonal and meridional transport by RCLVs and SSH eddies (we assume temporarily that SSH eddies are materially coherent). The eddy transport patterns of the two directions based on the SSH boundary are quite similar to the results based on the potential vorticity boundary (see Figure 3 in Zhang et al., 2014) because these two methods are essentially the same. The westward mass transport in the subtropical region and the eastward transport in the ACC region are remarkable (Figure 9b), with the meridionally integrated zonal transport reaching 30–40 Sv as well. However, the estimate based on RCLVs shows different patterns of zonal transport in the northeast Pacific and the tropical regions because of RCLVs moving eastward shown in Figure 8. The eastward eddy transports in these two regions are also captured by Xia et al. (2022) based on numerical model outputs. The peak value of meridionally-integrated zonal transport by RCLVs is only about 5 Sv, nearly an order of magnitude smaller than the transport by SSH eddies.

The huge overestimate of eddy coherent transport under the Eulerian framework can be attributed to two potential reasons. First, the material boundary of eddies is not defined appropriately using a contour from the instantaneous flow field. Previous studies (e.g., Beron-Vera et al., 2013; Liu et al., 2019) have shown clearly that the water exchange across the Eulerian eddy boundary is very active during the eddy lifetime and the Eulerian eddy size is usually larger than the real coherent core. Second, the period for which Eulerian eddies can maintain coherency is overrated. The eddy census of Pegliasco et al. (2022) identifies more than 2000 SSH eddies with lifetimes longer than 270 days in the Eastern Pacific, but Abernathey and Haller (2018) suggest that almost no coherent eddies can live that long. Our estimate serves as a reminder that the actual coherent eddy transport might be far smaller than the appealing results based on Eulerian methods.

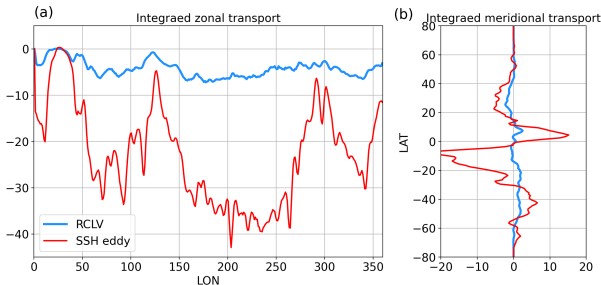

**Figure 10.** (a) Meridionally integrated zonal transport (in Sv) and (b) zonally integrated meridional transport by RCLVs (blue lines) and SSH eddies (red lines).

## 3.4 Dataset validation

In this subsection, Argo floats are used to verify the existence of identified coherent eddies and their ability to trap water parcels. Since its initial deployment in the early 2000s, the Argo profiling float array has expanded to encompass all the world's oceans with more than 3000 active floats. The standard Argo float is designed to conduct a 10-day cycle, during which it measures conductivity, temperature, and pressure at various depths. At the start of one cycle, the float sinks to a parking depth of 1000 m and drifts along with ocean currents for about 9 days. Then, it descends to a depth of 2000 m and rises to the surface while continuously measuring the ocean's properties. Upon reaching the surface, the float transmits its data to satellites before starting another cycle. The Argo data is available from the global data centers (ftp://ftp.ifremer.fr/ifremer/argo) and the position information of floats from 2000 to 2019 is considered here.

Due to the horizontal structure of mesoscale eddies being basically independent of the depth (Zhang et al., 2013), despite the eddy intensity having the vertical variability (Dilmahamod et al., 2018), it is believed that coherent eddies defined from surface geostrophic currents can maintain the coherent structure above a certain depth. Numerical simulations show that the averaged depth of coherent eddies does not exceed 500 m (Xia et al., 2022). However, the composite analysis based on Argo floats suggests that the density anomaly of mesoscale eddies in the west of Australia can penetrate deeper than 1000 m (see Figure 2 in He et al., 2021), which surpasses the parking depth of Argo floats. And there are many long-lived RCLVs near this area (Figure 6). Therefore, we search for Argo floats that are initially trapped by 90-day and 180-day RCLVs from GLED v1.0 in the southern Indian Ocean ($20 - 60°$S, $45 - 145°$E, black box in Figure 6c) and examine if these floats can be carried for a long range. A larger region than that in He et al. (2021) is used here in order to expand the sample size.

Figures 11a-11c show a case in which an Argo float is trapped by a 180-day RCLV for its entire lifespan. At the initial time, the float is located close to the eddy center and moves westward along with the eddy for more than 400 km. We then perform statistical analysis for 1001 90-day RCLVs and 270 180-day RCLVs. The calculation of the time-based distance between the eddy centers and trapped Argo floats is carried out. Figures 11d and 11e show that the majority of the floats can be continuously carried by RCLVs, with only a small portion escaping rapidly. The final distribution probability of Argo floats within a specified distance normalized by eddy radius is calculated. It is observed that for both 90-day and 180-day RCLVs, approximately 70%

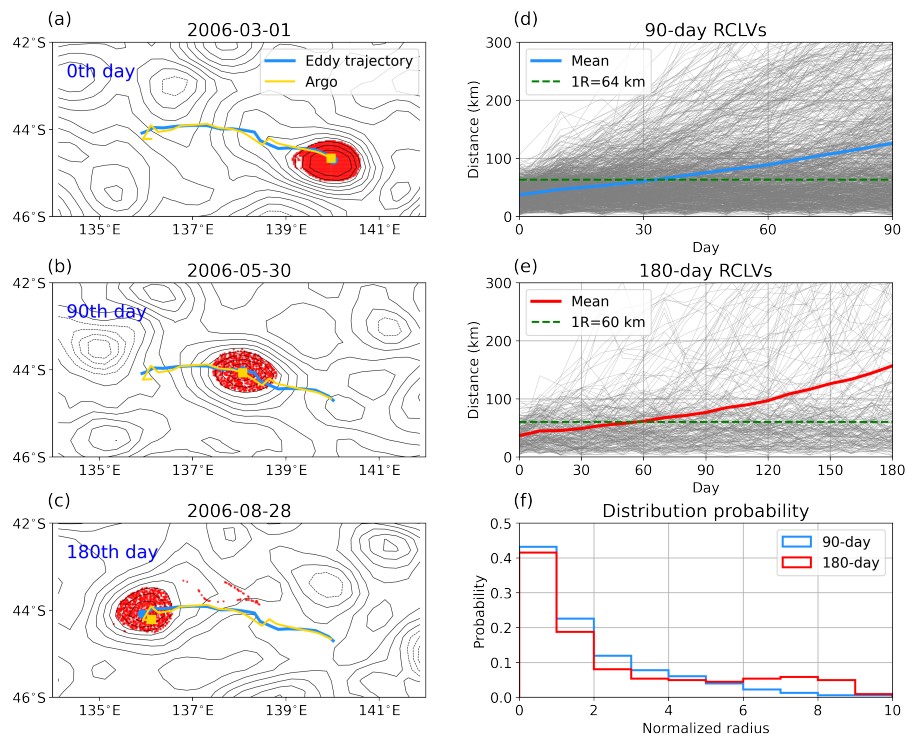

**Figure 11.** (a, b, c) A case showing that an Argo float is trapped by a 180-day RCLV (red dots), with the SLA field overlaid in black contours. The blue line tracks the eddy center's trajectory, while the yellow line tracks the Argo float's trajectory. (d, e) The distance between eddy centers and Argo floats that are initially trapped during the eddy lifetime for 90-day and 180-day RCLVs. Thin gray lines indicate the ensemble and the thick blue (red) line is the mean of 90-day (180-day) RCLVs. The green dashed line is the average radius of RCLVs. (f) The distribution probability of Argo floats with respect to the normalized eddy radius at the final time for 90-day (blue line) and 180-day (red line) RCLVs.

of the floats are still located within 3 times the eddy radius. Given the vertical tilt of mesoscale eddies (Li et al., 2022a), it is reasonable to conclude that these floats have been moving with coherent eddies since their initial trapping. The reason that some Argo floats escape from RCLVs might be because these eddies do not reach the parking depth of Argo floats. It is speculated that there would have been a stronger correlation if Argo floats were deployed at shallower depths. In general, these findings suggest that coherent eddies from GLED v1.0 exist in the real ocean and have the ability to transport water parcels.

## 4 Code and datasets availablility

Our dataset GLED v1.0 is available at https://doi.org/10.5281/zenodo.7349753 (Liu and Abernathey, 2022). It is convenient to load the data using Python, Matlab, or other programming languages. Detailed examples for reading and analyzing the data using Python can be found in a GitHub repository (https://github.com/liutongya/GLED), in which we also provide the related

algorithms to reproduce the generation of GLED v1.0. Users can apply these algorithms to regional or global identification of
coherent eddies with different lifespans based on velocity fields from observations or numerical simulations.

## 5   Conclusions and discussion

Methods employed to identify oceanic mesoscale eddies can be classified into Eulerian and Lagrangian frameworks, and nearly
all public global eddy dataset are based on the Eulerian framework (e.g., CS11) because of its operational simplicity. Eulerian
eddies are generally treated as coherent structures that can transport tracers such as heat, salt, and nutrients, and they have
been used widely to evaluate the material transport by eddies (e.g., Zhang et al., 2014), but recent studies under the Lagrangian
framework have provided clear evidence that (i) Eulerian eddies are far from being coherent studies and (ii) using Eulerian
methods will greatly overestimate the degree of real coherent transport (e.g., Abernathey and Haller, 2018; Liu et al., 2019). To
provide an additional option for oceanographers in studying mesoscale eddies, in this study, we proposed a global Lagrangian
eddy dataset (GLED v1.0) based on satellite observations.
Millions of Lagrangian particles with a resolution of $1/32°$ were advected by satellite-derived surface geostrophic velocities
for 180 days from the first day of every month over the period from January 1993 to June 2019. Using the LAVD method
proposed by Haller et al. (2016), we identified coherent eddies (RCLVs) with lifetimes of 30, 90, and 180 days to generate
GLED v1.0. This open-source dataset contains not only general features of coherent eddies (center position, equivalent radius,
rotation property, etc.), but also the trajectories of particles trapped by coherent eddy boundaries over the lifetime. To the best
of our knowledge, this is the first attempt to date to provide the position of Lagrangian particles advected by geostrophic flows
in an eddy dataset.
We compared the statistical features of RCLVs in GLED v1.0 with those of SSH eddies in META3.1exp. Unlike SSH eddies
that are broadly distributed in the global ocean basins, RCLVs tend to be generated close to the eastern boundaries, and the
RCLV numbers along the main paths of western-boundary currents and the ACC are very limited. The zonal and meridional
propagation speeds of RCLVs are found to be qualitatively similar to those of SSH eddies in most regions, but RCLVs are
much smaller than SSH eddies with a radius ratio of about 0.5. Using a fixed eddy depth, we calculated the mass transport by
RCLVs and SSH eddies. It was found that the zonal transport by SSH eddies can reach about 30–40 Sv, consistent with the
PV-based estimate of Zhang et al. (2014), but the transport by RCLVs is only about 5 Sv, nearly an order of magnitude smaller
than the Eulerian estimate. In addition, we conducted the dataset validation based on Argo floats and found that about 70% of
Argo floats that are initially trapped can always be carried by 90-day and 180-day RCLVs during the lifetime.
Although the estimated coherent eddy transport is quite weak, it does not mean that the role of mesoscale eddies in the
material transport is insignificant. Our primary point is the contribution of coherent structures to the total eddy transport
is limited, and the incoherent motions such as stirring and filamentation on the periphery of mesoscale eddies might make a
leading-order contribution (Hausmann and Czaja, 2012; Abernathey and Haller, 2018). More attention is required to understand
material transport by the filamentary structures, and the global particle trajectories produced by this study might be effective
for studying the motion behaviour outside coherent cores.

Because of the computation and storage pressures, GLED v1.0 only provides RCLVs identified over three time intervals. And it is still unclear how to reconcile the difference between the free Eulerian lifetime and the fixed Lagrangian lifetime. In order to better satisfy the users' needs, as well as the eddy information in the dataset, we provide the related algorithms to reproduce our results completely, from driving Lagrangian particles to RCLV identification. Users should feel free to modify the configuration (e.g., the date of releasing particles and the identification time interval) according to their own research.

Although we have produced a useful eddy dataset under the Lagrangian framework, one should note that not all studies must use Lagrangian eddies. Eulerian eddies are still convenient and meaningful when the coherent structure is not the main concern. Researchers should select the suitable method and dataset based on their objectives. This present study offers relief from the dilemma that the Eulerian eddy dataset is nearly the only option for studying mesoscale eddies.

One limitation of the present dataset is that RCLVs are based on surface geostrophic velocities, which might introduce potential errors due to limited spatial and temporal sampling of satellite data. Lacorata et al. (2019) evaluated the Lagrangian simulations based on satellite-derived currents with respect to real drifter trajectories. They found that surface currents from satellite observations have overall good Lagrangian skills for large-scale transport and dispersion numerical simulations, but the finite-resolution flow field tends to underestimate relative dispersion at scales smaller than 100 km. The differences between simulated and real drifter trajectories might come from the fact that geostrophic currents fail to capture small-scale processes and vertical motions, such as submesoscale currents and inertia-gravity waves. Since the present study also used satellite data to drive particles, we did not expect to see better performance in the direct comparison between simulated particles and real near-surface drifters. This is the reason why we verify the accuracy of the dataset using Argo floats, whose motions are mainly determined by geostrophic flows. In addition, Sinha et al. (2019) investigated particle evolution driven by hourly-, daily-, and weekly-averaged velocities from a $1/48°$ numerical simulation, and several cases showed that small-scale/high-frequency motions from hourly and daily velocities can make the coherent structure identified from weekly velocities leaky and cause strong vertical motions of particles. Recent works highlighted the role of small motions in material transport, but the extent to which these motions affect coherent structures is still an open question.

Here, we propose several potential application scenarios of GLED v1.0. First, it can be used to understand the structure and physical dynamics around mesoscale coherent eddies, including their interactions with multi-scale oceanic and atmospheric processes. Second, it can be used to estimate the coherent eddy transport of heat, salt, and nutrients, which can provide more accurate parameterization in climate and ecosystem models. Third, it can be used to explore the behavior and distribution of marine organisms and how they are influenced by coherent eddies. In addition, we need to remind users to be careful when using the particle trajectory in regions where submesoscale processes are active.

Although limitations exist, the satellite-derived geostrophic flow field is still the only large-scale velocity observation that resolves mesoscale structures, and our dataset has been verified to be reasonable at the geostrophic scale. We see this study as an important step toward fully understanding the features of mesoscale coherent structures, and we expect to update this dataset to version 2.0 once the observational data from the Surface Water and Ocean Topography mission become available. It would be quite meaningful to explore differences between the two versions, which will lead to new insights regarding multi-scale interactions and more accurate parameterization of eddy transport in numerical models.

*Video supplement.*   The video supplement is available at https://vimeo.com/773609039.
*Author contributions.*   RA proposed the idea and launched this project. TL and RA developed the related algorithm. TL conducted the offline
particle advection and data analysis. TL organized the eddy dataset. TL and RA wrote the manuscript.
*Competing interests.*   The authors declare that they have no conflict of interest.
*Disclaimer.*   Publisher's note: Copernicus Publications remains neutral with regard to jurisdictional claims in published maps and institutional
affiliations.
*Acknowledgements.*   This project has been supported by the National Natural Science Foundation of China (42106008, 42227901). We thank
Nathaniel Tarshish, Anirban Sinha, Wenda Zhang, and Ci Zhang for their early involvement to push this project forward. We thank two
anonymous reviewers for their helpful and constructive comments.

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
