# Peer review of "A global Lagrangian eddy dataset based on satellite altimetry"

_Earth System Science Data, 2022_

## Author Comment (AC1)

**Responses to reviewers' comments**

Title: A global Lagrangian eddy dataset based on satellite altimetry

MS No.: ESSD-2022-411

We sincerely thank two reviewers for their careful reading, helpful comments, and constructive suggestions, which have significantly improved the manuscript. We have carefully considered all comments and made modifications to the manuscript. Below, we provide a point-by-point response to the comments, with the original comments in black and our responses in blue.

**Reviewer 1**

This paper reflects the authors' commendable effort to make available to a broad scientific community a new atlas of mesoscale eddy structures in the global ocean, with a comparison of some salient features with those inferred from another atlas accepted as a good reference on the subject.

The originality of the study lies in the Lagrangian approach chosen for the diagnosis of the coherent structures, with the integration of an impressive number of numerical particles into the geostrophic velocity field that is calculated from the horizontal slopes of the AVISO sea-level product available weekly on a ¼° geographic grid.

The comparison of statistics between the two atlases sheds interesting light on the capabilities of each to account for the global mesoscale ocean activity. This comparison is all the more relevant since the source of data is the same (sea-level topography), and the results differ almost exclusively in the methodological approach for the analysis of the coherent structures (Lagrangian in the present study; Eulerian for the atlas used for comparison).

The article is well constructed, and properly illustrated. It provides useful background information for anyone wishing to work with this new inventory of oceanic mesoscale structures in the future. I find no notable flaws in the achievement of this first objective. Particular care seems to have been taken with the diversity of parameters making it possible to extract partial information from the atlas, for example for the purposes of regional studies.

As the number of atlases of equivalent nature is not high, this one cannot be reproached for being a nth newcomer: by its very availability, this atlas is an asset for the community.

Rather, my main criticisms relate first to some shortcomings in the assumptions used to promote a Lagrangian approach in the constitution of the atlas, and second to the nature of the comparison made with the reference atlas. I believe that the article will gain in quality, and the new atlas will be better understood and used if the authors carefully address the following points.

Response: Thanks. We acknowledge the positive feedback and have made some modifications.

(1) Major comment #1

The current basis of calculation of the new atlas is the sea-level topography distributed by AVISO. By construction, the eddy structures inventoried are the result of an altimeter afflicted with a form of myopia for the finer scale structures, with a resulting aliasing on the reality of the diameters and movements surveyed for the mesoscale structures. It would be interesting for the authors to discuss in more detail how this myopia affects the Lagrangian analysis they conduct: indeed, instead of having velocities fields sampled at fine spatial and temporal scales, the method has to work with the degradation already acted out in the AVISO product. Despite the use of a Runge-Kutta type numerical scheme, it is certain that some temporal and spatial scales are lost forever, and are not taken into account in the calculated displacements.

Since the authors report some experience with their method when applied to modeled velocity fields (that are able to resolve oceanic variability at finer scales), it would be appropriate to propose a reasoned discussion on the quantification, even coarse, of the errors that result from the application of their Lagrangian diagnosis.

What is missing in this paper is indeed an estimate of the methodological error committed by applying a Lagrangian methodology, which is in essence able to work on a kinematic field of excellent spatiotemporal resolution, on velocity data degraded in time and space. First, based on a fine-scale regional modeling of a fully developed eddy activity, it would be interesting to know the unbiased result of this method for identifying coherent structures and quantifying transport properties. Then, the modeled velocity field could be degraded (keeping only its geostrophic part, sampling it at a coarse weekly scale, extrapolating it on a ¼° geographic grid, etc.) and the identification and quantification conducted again. If such a sensitivity study, or an equivalent, is already present in the bibliography currently associated with the paper, it should be mentioned more explicitly.

Response: Thanks for your comments. We fully agree with your points and have made some modifications to the paper accordingly. In summary, while we certainly acknowledge the limitation of the altimetry data, we feel strongly that it is important to do the best we can with the available data, rather than simply wait for perfect data to appear. We will make the limitations clear and let the reader judge whether our product is useful for their specific application.

We choose to use the geostrophic flow field derived from satellite altimetry because this dataset is the only global-scale velocity observation that can resolve mesoscale structures. Another reason is that this study mainly focuses on 2D coherent eddies since the prevailing assumption considers mesoscale eddies to be approximately in geostrophic balance. Of course, geostrophic currents from satellite data cannot resolve small-scale/high-frequency structures due to the limited spatial (1/4 degree) and temporal (daily) sampling. We have mentioned this point several times to remind readers. Please see lines 93-99, 118-120, 134-135, and 384-385.

We have added some discussions to present that the potential errors of our analysis might come from the limitation of using satellite data. Lacorata et al. (2019) evaluated the Lagrangian simulations based on satellite-derived currents with respect to real drifter trajectories. They found that surface currents from satellite observations have overall good Lagrangian skills for large-scale transport and dispersion numerical simulations, but the finite-resolution flow field tends to underestimate relative dispersion at scales smaller than 100 km. We infer that the differences between simulated and real drifter trajectories are caused by some motions not identified by satellite data, such as submesoscale currents and inertia-gravity waves. In addition, Sinha et al. (2019) investigated the influence of small-scale/high-frequency motions on Lagrangian coherent structures based on a 1/48-degree numerical model output (MITgcm-llc4320). They applied hourly, daily, and weekly averaging to the velocity fields to filter out motions from different dynamical regimes, and they found that the coherent structure identified from weekly velocities would become leaky in experiments using hourly and daily velocities. Recent works have noticed the role of small-scale structures in material transport, but the extent to which these motions affect coherent structures is still an open question. Please see the modified discussion part in lines 384-409.

In addition, we added a new subsection 3.4 (lines 309-337) to present the accuracy of the dataset. Our results show that, given the vertical tilt of mesoscale eddies, about 70% of Argo floats that are initially trapped by coherent eddies can be carried by eddies for a long distance, which suggests that coherent eddies from GLED v1.0 exist in the real ocean and

have the ability to transport water parcels. Although limitations exist, our dataset is verified to be reasonable at the geostrophic scale.

Lacorata, G., Corrado, R., Falcini, F., & Santoleri, R. (2019). FSLE analysis and validation of Lagrangian simulations based on satellite-derived GlobCurrent velocity data. Remote sensing of environment, 221, 136-143.
Sinha, A., Balwada, D., Tarshish, N., & Abernathey, R. (2019). Modulation of lateral transport by submesoscale flows and inertia-gravity waves. Journal of Advances in Modeling Earth Systems, 11(4), 1039-1065.

(2) Major comment #2

I have serious reservations about the relevance of the estimates of the transport achieved at the global scale by coherent eddies. I understand that the authors wish to rely on calculations previously proposed in the literature, but I am not convinced that the definitions they use properly enlighten us on what this transport is, and help us appreciate thoroughly the differences between the two atlases.

Let's just focus on the use of the propagation velocity of an individual eddy, in the calculation of transport through the sections constituted by the meshes of a geographical grid of 1° resolution in latitude and longitude. With the adopted definition, two structures with identical geometric properties (same surfaces, same vertical extensions, therefore same volumes of water) will not contribute equally to the transport analysis if their mean propagation speeds (or longevities) differ. What are the physical implications of this definition? Can we agree that what truly matters is that a given volume of water (expressed in m3) is fully transmitted through a control section? Whether the transmission occurs in hours or weeks does not change the fact that the same volume transport must be inventoried at the longitude or latitude of control. Yet, in its current format and in my understanding, the definition implies that a faster or longer-lived eddy contributes more efficiently to local volume transport. If the authors intend to work with such a definition, meticulous care must be taken in presenting the meaning and utility, and in interpreting the calculated quantity.

Response: Thanks for your comments. We feel strongly that the rate of transport ($m^3/s$) and not simply the volume itself ($m^3$), is the most important quantity to focus on, because the transport of heat, freswater, nutrients, etc. is proportional to the volume transport rate. As an analogy, consider a different type of coherent structure: the gulf stream. No one would claim that it does not matter at what rate water is transferred from the tropical to subpolar ocean. The same is true for coherent eddy transport; all else being equal, a fasterpropagating eddy will move mass, heat, etc. more efficiently than its slower propagating counterpart. The approach used in this study has significant precedent in the literature.

By assuming no effective water exchange between the eddy interior and background flows, Zhang et al. (2014) and Dong et al. (2014) estimated the water volume, heat, and salinity transport on global scale. Although their results are shown to strongly overestimate the real coherent transport by eddies by neglecting the "leakiness" associated with their non-material eddies, the fundamental quantity they attempt to calculate (transport rates) is based on a sound definition grounded in the physics of transport. In this study, we choose this method for two reasons: (1) to demonstrate that the exaggerated zonal eddy transport (by Zhang et al. 2014) can be reproduced using SSH eddies and (2) to highlight the significant differences between SSH eddies and Lagrangian eddies in terms of coherence.

For each eddy, the instantaneous zonal transport can be expressed as $V_e C_x$, where $V_e$ is the eddy volume and $C_x$ is the zonal propagation speed. We can do this calculation for each SSH and Lagrangian eddies. We integrate all eddy snapshots (daily) over the studying period (1993-2019) to obtain the total transport $\sum V_e C_x$ within a $1° \times 1°$ bin. Dividing it by the number of satellite snapshots N and the length of one longitude degree L$_x$, we can get the physical quantity $Q_x = \sum V_e C_x / N L_x$ (in Sverdrup) that indicates the average zonal transport across the latitude section for each grid. The meridional transport can be estimated in the same way. Sorry for the typing mistake in the previous formula (4). In the calculation, we did not multiply the eddy lifetime since $\sum V_e C_x$ contains all eddy snapshots within each bin from 1993 to 2019. We have modified the text in lines 277-290 to present the physical implications of the calculated transport.

*Zhang, Z., Wang, W., & Qiu, B. (2014). Oceanic mass transport by mesoscale eddies. Science, 345(6194), 322-324.*

*Dong, C., McWilliams, J. C., Liu, Y., & Chen, D. (2014). Global heat and salt transports by eddy movement. Nature communications, 5(1), 3294.*

Other specific comments

(3) L21-22. The introduction could give some elements justifying the relevance of a better knowledge of mesoscale oceanic eddies also for the study of marine ecosystems.

Response: We have added two references to show the impact of oceanic eddies on marine ecosystems in lines 19 and 23.

*Mahadevan, A. (2016). The impact of submesoscale physics on primary productivity of plankton. Annual review of marine science, 8, 161-184.*

*McGillicuddy Jr, D. J. (2016). Mechanisms of physical-biological-biogeochemical interaction at the oceanic mesoscale. Annual Review of Marine Science, 8, 125-159.*

(4) L34-35. Even with the availability of continuous images of flow fields, an Eulerian, accurate identification of eddy centers and boundaries may remain a challenge due to the complexity of interlocking dynamical scales.

Response: We have modified this sentence in lines 35-37. "…if continuous images of flow fields are available, then searching for eddy centers and boundaries becomes relatively straightforward once the dynamical criterion is determined."

(5) L55-56. Observational studies are now focusing on exchanges along the vertical of an eddy structure, without being limited to interactions between the center of the eddy and its periphery.

Response: We agree with your comments that many studies are now focusing on the vertical motions around mesoscale eddies. The prevailing assumption is that mesoscale eddies are approximately in geostrophic balance, so this study mainly concentrates on two-dimensional coherent eddies based on geostrophic currents (lines 95-97). In lines 55-56, we are mainly discussing the lateral transport induced by eddies.

(6) L98-99. This is an important limitation of the identification, although legitimate for a global atlas. However, the paper should remind us that true oceanic eddies are not circular, may have subsurface properties more pronounced than their surface signature, have a complex organization along the vertical, may not be in solid-body rotation, are sensitive to ageostrophic motions, etc.

Response: We have mentioned that the vertical structure of mesoscale eddies is complicated and still an open issue (lines 93-99). In the modified discussion, we emphasized how the unresolved motion affects coherent eddies (lines 393-397).

(7) L108-109. The applied correction results in closed contours for trajectories within coherent structures considered as stationary. Again, the counterpart of the assumption is the omission of the vertical dynamics that exist within oceanic eddies. This point should be emphasized in the text.

Response: Please see the response to comment (1). We have highlighted this point in lines 93-99, 118-120, and 318-320.

(8) L115-116. This intensive seeding strategy requires discussion in light of the temporal and spatial scales already lost in the sea-level product distributed by AVISO (see main comment #1 above).

Response: We have added the text to explain why the high-resolution particle is adopted in lines 132-135. "It is important to note that using ultrahigh-resolution particles does not necessarily improve the resolution of the flow field as geostrophic currents are inherently unable to resolve small-scale/high-frequency processes, such as submesoscale flows, tides, and inertia-gravity waves. The real benefit is to avoid the discontinuous areas in the LAVD fields induced by coarse particle seeding and then to obtain the clear structure of coherent mesoscale eddies."

(9) L130-131. More care could be given to the justification of a monthly sampling of the initial positions chosen extremely loosely compared to the spatial sampling. What relationships link the two samplings, and how can these relationships be used to optimize the number of particle trajectories to be integrated?

Response: As we explained in the last comment, the intensive spatial seeding is to avoid the discontinuous areas in the LAVD images. Sensitivity tests by Abernathey and Haller (2018) suggest that a spacing of 1/32 degree is necessary for flow fields with a 1/4-degree resolution (lines 127-128). Actually, the spatial sampling is independent of the time interval of particle seeding. The particle is initialized every month because the shortest lifespan of eddies is generally considered as ~30 days. Due to computational and storage pressures, we cannot afford a higher frequency of particle seeding.

(10) L138-139. It should be reminded here that the coherent structures studied are, by construction of the method, idealized two-dimensional eddies.

Response: We have modified the text in lines 151 and 154. "Consider a two-dimensional coherent eddy …". "In a two-dimensional flow,…".

(11) L168-169. One may regret the empiricism of the choice of this combination of parameters. Chelton was often criticized for having too many control parameters in his eddy identification method. In the interest of fairness, this new paper could present a summary of all the adjustable parameters used for the derivation of a new eddy atlas.

Response: For eddy detection using the LAVD method, only two adjustable parameters are employed. Once a local LAVD maximum is determined, we search outward for closed LAVD curves. There might be multiple closed contours around a center, so two parameters

(CD and CI) are introduced to confine the boundary choice. We have explained the meaning of the two parameters and how the boundary change with parameter values in lines 166-175. The combination of CD<0.1 and CI>-1 is adopted according to the sensitivity analysis by Tarshish et al. (2018). Their summary indicates the parameter combination used here is a moderate threshold for identifying coherent eddies (lines 176-177 and 181-182). We also mentioned that "users should feel to modify the configuration according to their own research" in lines 378-379.

*Tarshish, N., Abernathey, R., Zhang, C., Dufour, C. O., Frenger, I., & Griffies, S. M. (2018). Identifying Lagrangian coherent vortices in a mesoscale ocean model. Ocean Modelling, 130, 15-28.*

(12) Figure 5. Avoid the use of longitudes larger than 180° (please express them in °W). In the caption, the notion of subtitle more likely refers to the title of each panel.
Response: We have changed the longitude label and the caption in Figure 5 and Figure7.

(13) L221-222. At the time of my review, the web page provided for the AVISO product unfortunately does not exist.
Response: We have tested that this web page works well (on 02/25/2023). There might be some network problems when you visited it. Please copy the URL instead of clicking the hyperlink. (https://www.aviso.altimetry.fr/en/data/products/value-added-products/global-mesoscale-eddy-trajectory-product.html)

(14) L270. Pointing out that the vertical structure of eddies is far from known and understood is a useful reminder, but should echo equivalent information to be discussed in the body of the introduction.
Response: We have added the text in the introduction. Please see lines 93-95 and 318-324.

(15) L280 and following lines, and possibly some parts of the conclusion. These passages should be carefully reconsidered once the definition of eddy transport has been unambiguously established (see main comment #2 above).
Response: Please see the comment (2). We have modified the unclear description of the eddy transport. The related results remain valid. And we highlight that "Our estimate serves as a reminder that the actual coherent eddy transport is far smaller than the appealing results based on Eulerian methods" in lines 307-308.

**Reviewer 2**

The Authors present a new global Lagrangian eddy dataset (GLED) containing information about various parameters like size, position, rotation, and lifetime of these coherent structures as well as trajectory data of numerical particles trapped inside these eddies. All quantitative information present in this database comes from massive numerical simulations of Lagrangian trajectories integrated from satellite-derived ocean surface current fields, covering a period of almost three decades.

I appreciate the remarkable effort made by the Authors to carry out a so challenging task although, personally, I have some doubts about the robustness of the results, for the reasons I will try to explain.

The "Lagrangian" eddy database, introduced in this work, is discussed by the Authors in comparison to pre-existing "Eulerian" eddy datasets, and the main differences between the two approaches are described with arguments that I agree with.

Response: Thanks. We acknowledge the positive feedback and have added the necessary validation to reduce your concerns.

(1) Line 7 (Abstract): "… but also the trajectories of particles…"

Personally, I do not like to consider Lagrangian trajectory simulations a "product" to store in a database as happens for other kinds of data, e.g., real ocean drifter trajectories or ocean current fields. One reason is that Lagrangian trajectories are sensitive to initial conditions and to the resolution of the velocity fields. But we will come back to this point later.

Response: We understand your perspective and would like to offer three points of explanation.

(1) Lagrangian trajectory simulations are indeed sensitive to various factors such as initial conditions and forcing velocities. To highlight the limitation of satellite data, we introduced the work of Lacorata et al. (2019) in the discussion. Moreover, we have included a new subsection 3.4 (lines 309-337) to present the validation of our dataset, which confirms the existence of coherent eddies in GLED v1.0 in the real ocean and their ability to transport water parcels.

(2) As mentioned in the introduction, it is worth noting that nearly all public global eddy datasets are based on the Eulerian framework, since identifying Lagrangian eddy over a global scale is a challenging task. So, a Lagrangian eddy dataset based on observations is needed to overcome the shortcomings of Eulerian methods and to provide an additional option for oceanographers studying mesoscale eddies. Given the difficulty and computational cost of these calculations, we think it is valuable to share this product with the scientific community, despite the noted limitations, so that other scientists may use the trajectory data for their own studies.

(3) While the geostrophic flow field derived from satellite data fails to capture small-scale/high-frequency motions, it remains the only global-scale velocity observation that can resolve mesoscale structures. And the Lagrangian product based on geostrophic velocities, such as the FSLE product presented by AVISO, has been widely accepted by the oceanographic community. Therefore, it is reasonable to present a global Lagrangian eddy dataset based on satellite data.

FSLE product is available at:
(https://www.aviso.altimetry.fr/en/data/products/value-added-products/fsle-finite-size-lyapunov-exponents.html)

(2) Lines 112-122 (Particle advection): "The first step… they never move."

Since 2D geostrophic velocity fields are considered, because of the nature of the Eulerian data, all results relate to large scale advection on the ocean surface, and any extrapolation to smaller scales and/or to sub-surface ocean layers must be treated with caution.

A particle spacing of 1/32° is claimed to be necessary for a good resolution of the RCLVs but it should be stressed that this does not mean to improve the resolution of the dynamics but only the definition of the large-scale features of the flow (>> 1/4°).

Response: We agree that the extrapolation to smaller scales and to sub-surface ocean should be carefully treated. We have added the related description in lines 132-135.

"It is important to note that using ultrahigh-resolution particles does not necessarily improve the resolution of the flow field as geostrophic currents are inherently unable to resolve small-scale/high-frequency processes, such as submesoscale flows, tides, and inertia-gravity waves. The real benefit is to avoid the discontinuous areas in the LAVD

fields induced by coarse particle seeding, which allows us to obtain the clear structure of mesoscale coherent eddies."

(3) Lines 138-184 (Lagrangian eddy identification): "For a coherent eddy … based on satellite observations."

The definition of LAVD fields and RCLV boundaries is supported by common sense arguments, but not much is said about the sensitivity of this eddy identification technique to the resolution of the velocity fields. For example, since the CI coefficient depends on the mean local particle separation, after a given time interval, it must be expected that, if the small scale velocity components, or part of them by means of sub grid parameterization techniques, were included in the trajectory evolution equations, the relative separation speed between particles would increase due to the general growth of the (Lagrangian) Lyapunov exponent with the smallest resolved scale.

Looking at figure 3, I wonder if the Authors have tried to make, at least, a qualitative comparison between a simulated coherent structure evolution and the behavior of a real ocean drifter, initially "trapped" inside the eddy. The Authors do not provide information about the accuracy of their Lagrangian trajectory simulations, but I think this problem should be at least mentioned in the text.

Response: We understand your concerns on the accuracy of this dataset. This comment is quite related to comments (6) and (7), so we put the main response here.

This study mainly focuses on 2D coherent eddies, as the prevailing assumption considers mesoscale eddies to be approximately in geostrophic balance. Both reviewers mentioned the limitation of the forcing velocities. We fully agree that geostrophic currents from satellite data cannot resolve small-scale/high-frequency structures due to the limited spatial (1/4 degree) and temporal (daily) sampling. We have mentioned this point several times to remind readers, including in lines 93-99, 118-120, and 132-135.

We introduced the results of Lacorata et al. (2019) to demonstrate the finite-resolution flow field from satellite data tends to underestimate relative dispersion at scales smaller than ~100 km. We infer that the differences between simulated and real drifter trajectories come from the structure not solved by geostrophic currents. Since our study also used the satellite data to drive particles, we did not expect to obtain better performance in the direct comparison between simulated particles and real oceanic drifters. So, we verified the

accuracy of our dataset using Argo floats, whose motions are mainly determined by geostrophic flows. Please see the discussion in lines 384-397.

We added a new subsection 3.4 (lines 309-337) to present the dataset validation. The composite analysis based on Argo floats suggests that the density anomaly of mesoscale eddies in the west of Australia can penetrate deeper than 1000 m (He et al., 2021), which surpasses the parking depth of Argo floats. And there are many long-loved RCLVs near this area. Therefore, we searched for Argo floats that are initially trapped by 90-day and 180-day RCLVs and examined if these floats can be carried for a long range.

Figures 11a-11c show a case in which an Argo float is trapped by a 180-day RCLV for its entire lifespan. The statistical results show the majority of the floats can be continuously carried by RCLVs, with only a small portion escaping rapidly. It is observed that for both 90-day and 180-day RCLVs, approximately 70% of the floats are still located within 3 times the eddy radius. Given the vertical tilt of mesoscale eddies (Li et al., 2022a), it is reasonable to conclude that these floats have been moving with coherent eddies since their initial trapping. These findings suggest that coherent eddies from GLED v1.0 exist in the real ocean and have the ability to transport water parcels.

[Figure]

Figure 11 (a, b, c) A case showing that an Argo float is trapped by a 180-day RCLV (red dots), with the SLA field overlaid in black contours. The blue line tracks the eddy center's trajectory, while the yellow line tracks the Argo float's trajectory. (d, e) The distance between eddy centers and Argo floats that are initially trapped during the eddy lifetime for 90-day and 180-day RCLVs. Thin gray lines indicate the ensemble and the thick blue (red) line is the mean of 90-day (180-day) RCLVs. The green dashed line is the average radius of RCLVs. (f) The distribution probability of Argo floats with respect to the normalized eddy radius at the final time for 90-day (blue line) and 180-day (red line) RCLVs.

*Lacorata, G., Corrado, R., Falcini, F., & Santoleri, R. (2019). FSLE analysis and validation of Lagrangian simulations based on satellite-derived GlobCurrent velocity data. Remote sensing of environment, 221, 136-143.*

*He, Y., Feng, M., Xie, J., He, Q., Liu, J., Xu, J., ... & Cai, S. (2021). Revisit the vertical structure of the eddies and eddy-induced transport in the Leeuwin Current system. Journal of Geophysical Research: Oceans, 126(4), e2020JC016556.*

*Li, H., Xu, F., & Wang, G. (2022). Global mapping of mesoscale eddy vertical tilt. Journal of Geophysical Research: Oceans, e2022JC019131.*

(4) Lines 219-260 (General features of global coherent eddies) "To assess GLED … along the eastern boundary"

Briefly, I find that all the differences between "Eulerian" and "Lagrangian" approach to eddy detection, here discussed, are plausible. Incidentally, there are many examples of kinematic velocity fields made of quasi stationary eddies advecting chaotic Lagrangian trajectories. So that, if the Eulerian field is analyzed, one finds out the existence of long-living coherent structures (i.e. with infinitely long Eulerian autocorrelation times) but with zero mean transport; on the other hand, the Lagrangian trajectories consist of aperiodic open pathways across the eddies and large-scale particle transport is due to chaotic diffusion.

Response: Thanks for your explanation. We also noticed the aperiodic Lagrangian coherent structures in cellular flows, as described by Lekien and Coulliette (2007), which can induce particle transport and chaotic stirring. Here is an interesting movie that illustrates this process (https://opendrift.github.io/gallery/example_double_gyre_LCS_particles.html).

It is worth noting that steady cellular flows are not commonly observed in the real ocean. In addition, as we mentioned earlier (in lines 21 and 66-67), coherent eddies are defined as structures that can trap the fluid within the eddy interior. This is obviously different from the aperiodic Lagrangian coherent structures.

*Lekien, F., & Coulliette, C. (2007). Chaotic stirring in quasi-turbulent flows. Philosophical Transactions of the Royal Society A: Mathematical, Physical and Engineering Sciences, 365(1861), 3061-3084.*

(5) Lines 261-288 (Global mass transport by coherent eddies) "One application … live that long."

Given that the 3D eddy structure is unknown, does it make sense to give quantitative information about the mass transport? Is the depth factor arbitrary or is there an argument to justify its value? Moreover, the eddy lifetime d in formula (4) could be seriously overrated with respect to the real ocean, for the reasons previously mentioned.

Response: We provide a quantitative transport by eddies for two reasons: (1) to demonstrate that the exaggerated zonal eddy transport (by Zhang et al. 2014) can be reproduced using SSH eddies and (2) to highlight the significant differences between SSH eddies and Lagrangian eddies. Due to the vertical structure of coherent eddies remaining an open issue, we use a constant depth following Abernathey and Haller (2018), which is supported by the findings of Roemmich and Gilson (2001) and Xia et al. (2022). This is a relatively crude estimate with accuracy only within an order of magnitude, and the focus here is mainly on the difference between the two types of eddies (lines 287-290). We also mentioned that the estimate serves as a reminder that the actual coherent eddy transport might be far smaller than the appealing results based on Eulerian methods (lines 307-308).

Sorry for the typing mistake in the previous formula (4). In the calculation, we did not multiply the eddy lifetime since $\sum V_e C_x$ contains all eddy snapshots within each bin from 1993 to 2019. Please see the modified text in Lines 276-284.

Abernathey, R., & Haller, G. (2018). Transport by Lagrangian vortices in the eastern Pacific. Journal of Physical Oceanography, 48(3), 667-685.

Roemmich, D., & Gilson, J. (2001). Eddy transport of heat and thermocline waters in the North Pacific: A key to interannual/decadal climate variability?. Journal of Physical Oceanography, 31(3), 675-687.

Xia, Q., Li, G., & Dong, C. (2022). Global oceanic mass transport by coherent eddies. Journal of Physical Oceanography, 52(6), 1111-1132.

Zhang, Z., Wang, W., & Qiu, B. (2014). Oceanic mass transport by mesoscale eddies. Science, 345(6194), 322-324.

(6) Lines 308-310 "To the best of … from physics to biology"
Lines 330-333 "Although we have produced … studying mesoscale eddies."

Personally, I find the presumed impact of this type of databases to the research activity of others is a bit overstated. While the existence of eddies is out of question, it is worth stressing that the assessment of the accuracy of the numerical simulations in reproducing the actual Lagrangian properties of real ocean tracers is a big open issue.

Response: We certainly acknowledge the limitation of the altimetry data, we feel strongly that it is important to do the best we can with the available data, rather than simply wait for perfect data to appear. We attempt to make the limitations clear and let the reader judge whether our product is useful for their specific application. As you suggested, we have shown the validation of the dataset based on Argo floats in the subsection 3.4. We modified the original text and stressed the Lagrangian particles are advected by geostrophic flows in lines 357-359.

(7) Lines 334-338 "One limitation … available"

While waiting for the next oceanographic missions, to update the database, I would suggest also to consider a Lagrangian validation of the numerical trajectory dataset (see, for example, Lacorata et al., 2019, FSLE analysis and validation of Lagrangian simulations based on satellite derived GlobCurrent velocity data, Remote Sensing of the Environment). Nobody expects the simulations to perfectly agree with the observations, but it is important to outline the limits of accuracy prior to any kind of application. I think that this could be an interesting information to potential end users of the product.

Response: Please see our response to comments (2) and (6). We rewrote the discussion part in lines 384-409. Several potential application scenarios of the dataset are proposed. And we remind users to be careful when using the particle trajectory in regions where submesoscale processes are active (lines 402-403).

---

## Author Response (AR2)

**Responses to reviewers' comments**

Title: A global Lagrangian eddy dataset based on satellite altimetry

MS No.: ESSD-2022-411

We sincerely thank three reviewers for their careful reading, helpful comments, and constructive suggestions, which have significantly improved the manuscript. We have carefully considered all comments and made modifications to the manuscript. Below, we provide a point-by-point response to the comments, with the original comments in black and our responses in blue.

**Reviewer 1**

I am satisfied with the way the authors have handled my major comment #1 and all my detailed remarks. Thanks to their new contributions or clarifications, the manuscript seems to me to have gained in quality.

On the other hand, I remain largely unsatisfied with their response to my major comment #2 on the notion of transport achieved by the coherent eddies identified by the atlas. In the current state of the revision process (response to my comment and content of the revised manuscript), I am compelled to ask for a new major revision, on this precise point.

The authors misinterpret the calculation of the transport associated with the movement of an eddy. Contrary to what they wrote in response to my comment, what matters is the added volumes of the eddies transferred across a control longitude (or latitude) segment over a given period, whatever their travel speed.

And no, a faster propagating eddy will NOT move mass, heat, etc. MORE EFFICIENTLY than its slower propagating counterpart. Efficiency here would be measured more in terms of an 'eddy gun': it is the number of such eddies crossing a reference line over a given period of time that matters, not their individual travel speed. Efficiency would also be measured in terms of porosity and property exchange with the neighboring ocean, rather than in terms of the speed at which the eddy moves.

With the formalism proposed by the authors, it is a bit like measuring the leakage of a tap by providing the individual momentum of the water drops (their volume multiplied by their velocity) rather than just their volume and number.

The transport approach is legitimate for qualifying permanent flows, but appears much less suitable for documenting sporadic flows such as those produced by the propagation of individual eddies, except after averaging over a given period when such eddies may follow one another. In the case of an isolated eddy, the information on its propagation speed is interesting in itself, but it does not combine well with the volume of the eddy to derive a transport index.

In the method used for the study, multiplication by the travel speed of the eddies is actually used to normalize the diagnosis by the number of times the same eddy is present in the same geographical grid cell, in order to avoid counting the passage of this eddy several times. In other words, the multiplication of the volume of the eddy by its travel speed is only made necessary by the nature of the counting carried out, i.e., the blind use of successive maps of eddies without consideration for the dynamic correspondence (same eddies that moved to different locations in the same grid cell) that exists between these maps.

One concern with the proposed calculation is that the travel speed of an eddy may not be constant in time and thus in space. This variability, possibly induced by the interaction of the eddy with its dynamic environment (such as other mesoscale structures) or geographical environment (bottom topography, coastline, etc.), introduces an error in the diagnosis which is not quantified in the study.

Finally, in my mind, the notion of local instantaneous transport associated with an eddy could ultimately refer to the volume of the eddy divided by the time it needs to completely cross a given control line, i.e. Ve Cx / (2 R), if the vortex is considered as a cylinder of diameter 2 R. This value can be maximized by simply considering the apparent cross-section of the vortex, in the direction transverse to the motion, in which case this transport is written as 2 R h * Cx (with h the eddy vertical extent). In contrast, the quantity Ve Cx introduced in line 278 has a physical meaning which is more of a momentum, but is certainly not transport.

Therefore, I ask that the revised version of the article be rigorous in its interpretation of the proposed transport. This is not the case with the current version, and the review process suggests that this may be due to a flawed understanding of the concept by the authors. Hence my initial request that meticulous care be taken in the presentation of the meaning and usefulness, as well as in the interpretation of the quantity calculated quantity.

Response: Thanks for your positive feedback on our revisions in the last round. Given that both Reviewer 3 and editor think the estimate of eddy transport is beyond the scope of ESSD, we decide to remove this part in the new manuscript. And we have changed the related text in Abstract, Section 3, and Section 5 (in red).

According to your comments, we provide our explanations here as a discussion.

Does the eddy propagation speed matter? Our answer is yes. We used this formula $Q_x = \sum V_e C_x / N L_x$ [Eq. 1] to estimate the zonal mass transport by eddies. [Eq. 1] represents the volumetric flow rate (m3/s or Sv) but rather the net volume (m3). Imagine two mesoscale eddies with the same size and a positive temperature anomaly. Both eddies move from region A to region B. There is no doubt that the faster eddy will more efficiently heat the water of region B than the slower eddy. This principle applies equally to transport of mass, salinity, or other tracers by eddies. The eddy propagation speed is an important factor for measuring the property exchange rate with the neighboring ocean.

The fundamental definition of eddy transport/flux for a tracer q is Q=v'q' [Eq. 2], where v' is velocity perturbation and q' is tracer anomaly. [Eq. 2] also suggests that the eddy speed does matter. But, this formula relies on high spatiotemporal resolution data. [Eq. 1] was proposed based on [Eq. 2], with $C_x$ (eddy propagation speed) being analogous to a velocity perturbation against the mean flow and $V_e$ representing the integrated tracer anomaly within an eddy. Dong et al. (2014) and Zhang et al. (2014) estimated the eddy transport using [Eq. 1] by assuming that mesoscale eddies can always trap the interior water, and they concluded that the eddy transport is mainly induced by the trapping of individual eddies (In other words, they suggested that [Eq.1] can largely represent [Eq.

2]). These two papers have been cited about 800 times in total, and [Eq. 1] has been adopted to calculate the eddy transport by more than 50 papers.

[Eq. 1] represents the integration of all instantaneous eddy transport, which is a statistical concept. As long as an eddy snapshot (daily) exists in a grid, it is considered to contribute to the total transport, and it is counted as a sample. The eddy propagation speed was calculated according to the daily eddy position, so the evolution and travel of eddies are considered. Note that the grid length $L_x$ in the denominator of [E1. 1], which means that $Q_x$ is roughly the averaged zonal transport across the latitude line for each grid. This physical meaning is reasonable, but the transport degree (e.g. more than 30 Sv by Zhang et al. 2014) is **strongly overestimated**! The core problem lies in whether these eddies are coherent structure, in other words, whether they can truly trap the interior water. This is precisely the topic discussed in our paper.

If you agree that the instantaneous transport can be expressed by Q=2RhCx [Eq. 3], then we share the same viewpoint. There is no fundamental difference between [Eq. 1] and [Eq. 3] for a statistical estimate since [Eq. 1] also represents the averaged transport across a certain line (after dividing by the grid length $L_x$). Our previous studies (Abernathey and Haller 2018; Liu et al. 2019) did used Q=2RhCx [Eq. 3] to estimate the eddy transport. In this study, we choose [Eq. 1] for two reasons that were clarified in our last response. First, we attempt to reproduce the exaggerated transport by Zhang et al. (2014) using their widely accepted method. Second, we attempt to highlight the significant **differences** between SSH eddies and Lagrangian eddies in terms of coherence (It's a fair comparison).

In our manuscript, we have emphasized that "This is a relatively crude estimate with accuracy within an order of magnitude, and the focus here is mainly on the difference between the two types of eddies". Our estimate serves as a reminder that the actual coherent eddy transport might be far smaller than the appealing results based on Eulerian methods. Since the accurate contribution of coherent transport by eddies remains unclear, we will organize another paper to continue this discussion in the near future.

*Zhang, Z., Wang, W., & Qiu, B. (2014). Oceanic mass transport by mesoscale eddies. Science, 345(6194), 322-324.*

*Dong, C., McWilliams, J. C., Liu, Y., & Chen, D. (2014). Global heat and salt transports by eddy movement. Nature communications, 5(1), 3294.*

*Abernathey, R., & Haller, G. (2018). Transport by Lagrangian vortices in the eastern Pacific. Journal of Physical Oceanography, 48(3), 667-685.*

*Liu, T., Abernathey, R., Sinha, A., & Chen, D. (2019). Quantifying Eulerian eddy leakiness in an idealized model. Journal of Geophysical Research: Oceans, 124(12), 8869-8886.*

**Reviewer 3**

The papers is presenting a global eddy data set obtained with a Lagrangian-averaged vorticity deviation (LAVD) method. The paper is well written and of general interest. The many methods to calculate eddy boundaries have been discussed and presented in the paper. The proposed LAVD method one of these method and the authors correctly state: We encourage users of our product to be mindful of the limitations of the underlying satellite-derived geostrophic velocity fields used to derive our coherent eddies.

The authors also state that the data set is an additional option to oceanographer in studying eddies interactions as well as physical – biochemical interactions.

The paper is very well divided in three parts: methodology, statistics and availability and discussion. I would say that the paper is at 90% coherent to ESSD requirements: ESSD understands original research data as data generated by observation of the Earth system. In most cases this excludes reprocessing, postprocessing, or reanalysis of model outputs (such as climate model outputs, environmental modeling outputs, output of classification algorithms, and so forth). [ESSD - Manuscript types (earth-system-science-data.net)]

In ESSD the authors are requested to publish data that can be re-used. Probably the paragraph 3.3. has been included in the manuscript to provide an example of application

and re-use. From my point of view this part should be left to another paper to be submitted to a more dedicated journal. My view is presented in the specific comment.

Specific comment

The authors calculate VeCx with Cx the propagation velocity. But while in an Eulerian system the position of a particle p in time can be written as $X(p,t)$, in a Lagrangian system the position of the particle must be expressed only in a moving system and therefore must be written as $X((p,t),t)$.

Transport quantities introduced in the paper need to be clarified on their meaning. There are many limitations in the methods that have been previously underlined by the authors: assumed geostrophy, unknown subsurface vertical features of eddies, regional variability, etc. I am not sure that the Cx calculated by the authors can be used for an estimation of transport, if limitations are not discussed and quantified. And in general I would agree with referee 1: in the specific case it is quite arbitrary to calculate this quantity, as a component of the transport.

Conclusion

The paper can be published if it is only a presentation of the data sets, without the discussion on transport. Paragraph 3.3 is posing many problems and questions, and (from my point of view) is outside the ESSD scope.

Response: Thanks for your positive feedback on our work. As you suggested, we have removed the estimate of eddy transport in the new manuscript. And we have changed the related text in Abstract, Section 3, and Section 5 (in red).

We agree that the specific value of eddy transport is still unclear. In our manuscript, we have emphasized that "This is a relatively crude estimate with accuracy within an order of magnitude, and the focus here is mainly on the difference between the two types of eddies". Our estimate can serve as a reminder that the actual coherent eddy transport might be far smaller than the appealing results based on Eulerian methods.

Regarding the detailed explanations of eddy transport (why the eddy propagation speed matters; the physical meaning), please see our response to Reviewer 1.

**Editor**

Public justification (visible to the public if the article is accepted and published):

In agreement with the referees comments I am convinced that this paper is well written and of general interest but, at this stage, it needs to be revised in section 3.3 where the definition of transport is questionable as argued by referee 1. Moreover this paragraph, as suggested by referee 3, can be considered outside the ESSD scope. See ESSD - Manuscript types (earth-system-science-data.net). Referee 3 suggests to produce a new version without the discussion on transport. This will overcome the debate with referee 1, leaving referee 2's suggestion as the only minor revision to be made.

Additional private note (visible to authors and reviewers only):

While referee #2 suggested to review only few details in the writing. Referees 1 and 3 indicated that the paper needs a major revision essentially for the same reason related to the definition the transport due to coherent eddies. In my opinion, their comments are reasonable and should be taken into consideration. In particular I agree with the suggestion of referee 3 about the opportunity to consider paragraph 3.3, that was the origin of the discussion with referee 1, not exactly fully within the scope of ESSD.

Response: Thanks for your positive feedback on our work. We have removed the estimate of eddy transport in the new manuscript since this part is beyond the ESSD scope. And we have changed the related text in Abstract, Section 3, and Section 5 (in red). In addition, you mentioned that Reviewer 2 proposed a few comments in the writing, but we did not receive the report in the system.